



# Stratospheric aerosol layer perturbation caused by the 2019 Raikoke and Ulawun eruptions and climate impact

Corinna Kloss[1], Gwenaël Berthet[1], Pasquale Sellitto[2], Felix Ploeger[3,4], Ghassan Taha[5,6], Mariam Tidiga[1], Maxim Eremenko[2], Adriana Bossolasco[1], Fabrice Jégou[1], Jean-Baptiste Renard[1], and Bernard Legras[7]

[1]Laboratoire de Physique et Chimie de l'Environnement et de l'Espace, CNRS/Université d'Orléans, UMR 7328, Orléans, France
[2]Laboratoire Interuniversitaire des Systèmes Atmosphériques, UMR CNRS 7583, Université Paris-Est Créteil, Université de Paris, Institut Pierre Simon Laplace (IPSL), Créteil, France
[3]Forschungszentrum Jülich GmbH, Institute of Energy and Climate Research (IEK-7), Jülich, Germany
[4]Institute for Atmospheric and Environmental Research, University of Wuppertal, Wuppertal, Germany
[5]Universities Space Research Association, Greenbelt, Maryland, USA
[6]NASA Goddard Space Flight Center, Greenbelt, Maryland, USA
[7]Laboratoire de Météorologie Dynamique, UMR CNRS 8539, ENS-PSL/ Sorbonne Université/ École Polytechnique, Paris, France

**Correspondence:** Corinna Kloss (corinna.kloss@cnrs-orleans.fr)

**Abstract.** In June 2019 a stratospheric moderate eruption occurred at Raikoke (48°N, 153°E). Satellite observations show the injection of ash and $SO_2$ into the lower stratosphere and an early entrainment of the plume into a cyclone. Following the Raikoke eruption stratospheric Aerosol Optical Depth (sAOD) values increased in the whole northern hemisphere and tropics and remained enhanced for more than one year, with peak values at 0.040 (shorter-wavelength visible, higher northern

5    latitudes) to 0.025 (shorter-wavelength visible, average northern hemisphere). Discrepancies between observations and models indicate that ash has played a role on evolution and sAOD values. Top of the atmosphere radiative forcings are estimated at values between -0.3 and -0.4 W/m$^2$ (clear-sky), and of -0.1 to -0.2 W/m$^2$ (all-sky), comparable to what was estimated for the Sarychev eruption in 2009. Almost simultaneously two significantly smaller stratospheric eruptions occurred at Ulawun (5°S, 151°E) in June and August. Aerosol enhancements from the Ulawun eruptions had mainly an impact on the tropics and

10   southern hemisphere. The Ulawun plume circled the Earth within one month in the tropics. Peak shorter-wavelength sAOD values at 0.01 are found in the tropics following the Ulawun eruptions, and a radiative forcing not exceeding -0.15 (clear-sky) and -0.05 (all-sky). Compared to the Canadian Fires (2017), Ambae eruption (2018), Ulawun (2019) and the Australian fires (2019/2020) highest sAOD values and RF are found for the Raikoke eruption.

## 1   Introduction

15   Severe volcanic eruptions can inject a significant amount of sulfur-containing species and, potentially, ash material, directly into the UTLS (Upper Troposphere-Lower Stratosphere). In the UTLS, secondary sulfate aerosols (SA) are formed by conversion of sulfur dioxide ($SO_2$) volcanic emissions to particles. Because of the limited potential of dry and wet deposition in the UTLS, these particles (SA, in particular, but also fine ash particles, when present) have a long lifetime. Additionally, SA are





reflective and effectively scatter short-wave radiation back to space, thus producing a net cooling effect on the climate (Kremser
et al., 2016). The extent of the impact on the global stratospheric composition and climate, from a volcanic eruption, depends
on various parameters: 1) chemical composition and concentration of the plume, 2) geographical location of the erupting vol-
cano, (3) injection altitude, (4) dynamical situation at the time and location of the injection. (1) The sulfur burden in the plume
determines the resulting SA formation and dominates the climate impact (Kremser et al., 2016). Whether the initial plume
contains ash or not can modify the chemical and micro-physical evolution pathways, aerosol formation/evolution and can alter
related dynamical features (radiative balance including local diabatic heating) (Robock, 2000; Vernier et al., 2016). (2) A trop-
ical volcano producing sulfate material into the UTLS usually has a larger geographical impact than a similar sized eruption at
higher latitudes. From the tropical lower stratosphere air masses have the potential to be transported at very long distance, in
both hemispheres and up to higher latitudes, within the Brewer-Dobson-circulation (BDC) (Butchart, 2014). (3) The aerosol
lifetime of a plume is also connected with the injection altitude relative to the tropopause. A higher injection altitude results in
a longer potential transport within the BDC, which leads to a longer potential lifetime of the formed or pre-existing aerosol.
(4) The dynamical situation around the plume (cyclones, anticyclones, jet etc.) can modify the transport pathways and, in some
cases, lead to a fast transport/distribution (Fairlie et al., 2014; Wu et al., 2017).

The Pinatubo ($15.13°N$, $120.35°E$) eruption in June 1991 is the most recent example of a volcanic eruption with a global
climate influence. Around 20 Tg of $SO_2$ have been injected into the lower stratosphere (Bluth et al., 1992), which caused a
global mean surface temperature drop of nearly $0.4°C$ (Thompson et al., 2009), although its amplitude has been debated and
revised (Canty et al., 2013; Wunderlich and Mitchell, 2017). Since then, no volcanic eruption with a comparable impact on the
climate occurred. However, even without major (Pinatubo-like) stratospheric eruptions it has been shown that, during the past
two decades, moderate eruptions substantially increased the amount of stratospheric aerosols (Vernier et al., 2011; Solomon
et al., 2011; Ridley et al., 2014). Some prominent 'moderate-sized' volcanic eruptions during the last decade were recorded,
in particular at Kasatochi on August $7^{th}$ 2008 in southwestern Alaska ($52.17°N$ and $175.51°E$), Sarychev on June $15^{th}$ 2009
on the Kuril Islands ($48.1°N$, $153.2°E$) and Nabro on June $12^{th}/13^{th}$2011 in the Afar Triangle between Ethiopia and southern
Eritrea ($13.37°N$ and $41.47°E$). The eruption at Kasatochi produced an initial $SO_2$ injection of 0.7-2.2 Tg (Günther et al.,
2018; Kristiansen et al., 2010; Krotkov et al., 2010). The $SO_2$ burden injected from the Sarychev eruption into the UTLS was
originally calculated at 1.2 $\pm0.2$Tg Haywood et al. (2010). After Pinatubo, the Nabro eruption was considered as the largest
single injection of $SO_2$ to the UTLS with 1.3-2 Tg (e.g., Clarisse et al., 2011; Sawamura et al., 2012).

An accurate description of such stratospheric volcanic eruptions is challenging. Fromm et al. (2014) raise awareness of unsat-
isfactory conditions in terms of data quality (for the example of OSIRIS satellite measurements), but also conflicting injection
sequence information used for potential model studies, which can lead to different conclusions of the same volcanic eruption.
Furthermore, for the Sarychev eruption several re-estimations during the past decade yield different numbers between 0.8 and
1.5 Tg for the injected $SO_2$ burden (Clarisse et al., 2012; Jégou et al., 2013; Höpfner et al., 2015; Günther et al., 2018), which
in itself indicates the complexity that goes along with a single eruption.

Ten years after the Sarychev eruption, in 2019 another eruption similar in location, time of the year and load of injected aerosol
precursors took place at Raikoke ($48°N$ and $153°E$) on June $21^{st}/22^{nd}$ 2019. At almost the same time the volcano at Ulawun





erupted on June $26^{th}$ and August $3^{rd}$ 2019 (5°S and 151°E) and two stratospheric fire events occurred in Canada, Alberta

(June) and Siberia (July).

This study aims at a first description of the complex situation in the UTLS around the Raikoke and Ulawun eruptions. We investigate the injection, global transport and climate impact of the 2019 eruptions at Raikoke and Ulawun.

In Section 2, we introduce the data sets, models and their respective set up. Section 3 gives an overview of both volcanoes and some key information on the presented eruptions. The early phase of the injected Raikoke plume and the global transport of

the Raikoke and Ulawun plumes are analyzed in Section 4 and the resulting climate impact is estimated in Section 5. Finally conclusions are drawn.

## 2 Methods

### 2.1 OMPS

The Ozone Mapping Profiler Suite Limb Profiler (OMPS-LP) flies onboard the Suomi National Polar-orbiting Partnership

satellite since October 2011. It was originally designed for height-resolved atmospheric ozone observations (Loughman et al., 2018; Bhartia and Torres, 2019). Aerosol extinction measurements at 675 nm are provided from 10 to 40 km altitude on a 1 km vertical grid. Three slits separated horizontally by 4.25° result in three measured profiles at each point in time separated by 250 km of the tangent points at the Earth's' surface. The vertical resolution is ∼1.6 km. Here, we use the aerosol extinction profile measurements from 2017 onwards of the NASA OMPS data product version 1.5. (Rault and Loughman, 2013). A near-

global coverage is produced within 3-4 days. Tropopause values are included in the data set from the MERRA-2 (Modern- Era Retrospective analysis for Research and Applications, Version 2)) forward processing. To avoid removing enhanced aerosol layers that were mistakenly identified as clouds, we use the unfiltered OMPS data set. With its high sampling rate, we use the OMPS data set to study the global transport of the respective volcanic plumes in the lower stratosphere.

### 2.2 SAGE III/ISS

As part of an ongoing instrumental series, a Stratospheric Aerosol and Gas Experiment instrument flies on board the International Space Station (SAGE III/ISS). It is a solar and lunar occultation instrument, providing, among other parameters, vertical profile observations of ozone, water vapor, nitrogen dioxide and nitrogen trioxide concentration, and aerosol extinction. Aerosol extinction values from the solar occultation measurements are provided for various wavelengths: 384, 449, 521, 676, 756, 869, 1020 nm. Measurements are provided since June 2017 between 60°S and 60°N on a 0.5 km vertical grid from

0.5 (or cloud top) to 40 km altitude. The vertical resolution is ∼1 km. Similar to OMPS, the tropopause information is included in the data set from the MERRA-2 reanalysis. We use the data version 5.1. Chen et al. (2019) find a good agreement between SAGE III/ISS and OMPS data. In particular, after the eruption at Ambae a small discrepancy (<+-10%) was found. As a solar occultation instrument, SAGE III/ISS provides 30 measurements per day. This relatively low sampling rate (e.g. compared to OMPS) limits the interpretability of the finer transport features analyzed with SAGE III/ISS. However, the better vertical



resolution and observations on multiple wavelengths compared to OMPS, are a marked added-value when spatio-temporally averaged data are used for the radiative forcing calculations.

## 2.3 Himawari

Himawari-8 is a geostationary satellite at 140°E from the Japanese Space Agency providing measurements of temperature, clouds, precipitation and aerosol distribution since 2015 (launched in 2014). It has an expected lifetime of 8 years and will be

replaced afterwards by Himawari-9. It observes the area of East Asia and the Western Pacific (Da, 2015). We use the brightness temperature (BT) observations from the 16-channel multispectral imager from the Clear Sky Radiance product (Uesawa, 2009). The data have a spatial resolution at sub-satellite point of 2 km for the infrared channels (0.46–13.3 $\mu$m). For the interpretation of the results in this study, we use the operational Eumetrain RGB recipes (Eumetrain, 2020), which allows to discriminate clouds, ash and $SO_2$, thanks to the combination of the infrared channels at 8.5, 10.4 and 12.3 $\mu$m. The Dust RGB product

performs better for volcanic plumes than the Ash RGB product at large viewing angles. Thus, the Dust RGB product is used to describe the first phases of dispersion of the Raikoke plume.

## 2.4 IASI

The Infrared Atmospheric Sounding Interferometer (IASI) is a Fourier transform spectrometer (Clerbaux et al., 2009), operating between 645 and 2760 cm$^{-1}$ (3.62 to 15.5$\mu$m) spectral range, on board the MetOp-A/B/C spacecrafts series since

2006/2012/2018, respectively. The instrument provides global coverage every 12 hours, thanks to its circular foot-prints of 12 km radius spaced by 25 km at nadir and a swath of 2200 km. The IASI has the relatively high apodized spectral resolution of 0.5 cm$^{-1}$. While its primary target is the monitoring of meteorological parameters (surface temperature, temperature, humidity profiles and cloud information), IASI also provides high-quality information on trace gases parameters and particles, including volcanic effluents (e.g., Clarisse et al., 2013; Carboni et al., 2016; Ventress et al., 2016; Guermazi et al., 2020).

In this work, we exploit the high spectral resolution of IASI to resolve one absorption line of $SO_2$. to provide a quick estimate of $SO_2$ detection in volcanic plumes (i.e. without the use of a detailed and computationally-demanding inversion algorithm, e.g. based on radiative transfer model-based spectral fitting). We define the following parameter:

$$D_{SO2} = R(\nu_2)/R(\nu_1) \tag{1}$$

R($\nu$) represents the radiance observed from IASI at the wavenumber $\nu$. The two values $\nu_1$= 1129.25 nm and $\nu_2$= 1130.25 nm,

represent two spectrally-close wavenumbers, the first at the center of a $SO_2$ absorption line and the second outside. Values larger than 1.0 of $D_{SO2}$ are linked to spectra where $SO_2$ is detected.

## 2.5 LOAC in situ measurements

The Light Optical Aerosol Counter (LOAC) is an Optical Particle Counter suitable for tropospheric and stratospheric observations of aerosol concentration (Renard et al., 2016). It is light and compact enough for in situ measurements using weather

balloons. It provides particles number concentrations for 19 sizes in the 0.2 – 50 $\mu$m size range, with an uncertainty of ±20%



for concentrations higher than 10 particles cm$^{-3}$; the uncertainty increases to about $\pm 30\%$ for submicronic particle concentrations higher than 1 particle cm$^{-3}$, and to about $\pm 60\%$ for concentrations smaller than $10^{-2}$ particle cm$^{-3}$. The raw LOAC concentrations are corrected in term of sampling efficiency for observations during balloon ascent (Renard et al., 2016), the sampling being dominated by sub-isokinetic conditions and the divergence of the flow field at the inlet entrance. LOAC V1.5

data used in this study have been improved in comparison with LOAC V1.2 presented in Renard et al. (2016), resulting in reduced stray light and higher signal-to-noise ratio using a more powerful laser source (65 mW instead of 25 mW formerly). The size distributions have been converted to 675 nm extinction using Mie scattering theory, assuming spherical particles with a refractive index corresponding to stratospheric sulfuric acid particles. Only size classes below 1 $\mu$m have been used to avoid spurious effects (i.e. local enhancements in the calculated extinction value) resulting from the transient presence of

micrometric particles. As a result, only a partial extinction have been derived. In this study, we use LOAC observations from France (Ury, 48.34°N, 2.60°E).: 11 observations, i.e. 22/3/2019, 8/8/2019, 29/8/2019, 16/9/2019, 11/10/2019, 30/10/2019, 20/11/2019, 3/12/2019, 7/1/2020, 6/2/2020, 6/3/2020. For the transformation from aerosol concentration to extinction for the Aerosol Optical Depth comparisons with satellites, only size classes below 1 $\mu$m are used because of artefacts appearing for size classes above 1 $\mu$m.

### 2.6 Transport simulation with CLaMS

The Chemical Lagrangian Model of the Stratosphere (CLaMS) is a Lagrangian Chemistry transport model. The model transport is based on 3D forward trajectories and an additional parameterization of small-scale mixing (McKenna et al., 2002; Pommrich et al., 2014). The transport is driven by the ERA-5 meteorological data (Hersbach et al., 2020). As CLaMS uses an isentropic vertical coordinate in the stratosphere, vertical transport in the model is driven with the reanalysis total diabatic heating rate.

Here, we perform CLaMS passive transport simulations for both volcanic eruptions. Chosen boxes in space and time are filled with a passive tracer and monitored in terms of dynamical behavior for the following months. The initialization box for Raikoke was chosen from $23^{rd}$-$24^{th}$ of June 2020, 163°E-170°W, 49-62°N and 335-460 K potential temperature. For the Raikoke eruption the box was chosen according to Hedelt et al. (2019). IASI/Metop-B data from Aeris (2018) show similar injection altitudes. The Ulawun transport was initialized from August $3^{rd}$ to $4^{th}$ 2019, 137-178°E, 10°S-5°N and 350-385 K

potential temperature, according to IASI/Metop-B data.
Note that the CLaMS model alaysis has certain limitations. As a consequence, from choosing a box shape for the initialization of the simulations, many of the presented trajectories do not exactly originate from the actual plume positions. However, we want to emphasize that the CLaMS simulations in this study are purely to be taken as a rough idea of the transport from the respective initialization boxes.

### 2.7 UVSPEC radiative forcing calculations

We use the UVSPEC (UltraViolet SPECtrum) radiative transfer model as implemented within the LibRadtran package (Mayer and Kylling, 2005) (http://www.libradtran.org/doku.php). With the UVSPEC the daily-average (equinox-equivalent) regional shortwave surface and top of the atmosphere (TOA) radiative forcing (RF) are estimated. The RF estimations are based on





radiation flux simulations in the spectral range from 300 to 3000 nm, with a 0.1 nm spectral resolution. The radiative transfer
equation is parameterized and solved as follows: - the solar flux spectra used to drive the simulations are taken from Kurucz
(Kurucz, 2005). - vertical profiles of temperature, pressure, humidity and gas concentration come from the climatological stan-
dards of the Air Force Geophysics Laboratory (AFGL). Mid latitude standard profiles are used for simulations of the Raikoke
plume, while tropical standard profiles are used for Ulawun. - The molecular absorption is parameterized with the LOWTRAN
band model (H. Pierluissi and S. Peng, 1985) (as adopted from the SBDART code). - The radiative transfer equation is then
solved with the SDISORT method (the pseudo-spherical approximation of the discrete ordinate method (DISORT)). The vol-
canically perturbed simulations are carried out by adding average SAGE III/ISS profile observations of the volcanic aerosol
extinction coefficient (details on the spatio-temporal identification of the volcanic perturbations are described in Sect. 4). As
baseline, SAGE III/ISS aerosol extinction profiles are taken for background conditions, i.e. without volcanic aerosols (details
on the spatio-temporal identification of the background are described in Sect. 4). For both setups (background and volcanically
perturbed) we carry out multiple runs with varying solar zenith angles (SZA). Finally, the daily-average shortwave TOA ra-
diative forcing is calculated by integrating the SZA-averaged upward diffuse irradiance for the background scenario over the
whole shortwave spectral range. The shortwave surface radiative forcing is calculated with the SZA-averaged downward global
irradiance with aerosols minus the background scenario, integrated over the whole spectral range.

## 2.8 WACCM model

Model simulations were performed using the global CESM1 (Community Earth System Model 1) using its Whole Atmosphere
Community Climate Model (WACCM) module linked to the CARMA (Community Aerosol and Radiation Model for Atmo-
spheres) module, involving the sulfur cycle with a sectional aerosol scheme (English et al., 2011). Land, sea ice, and rivers
were active modules, whereas oceans were prescribed. The spatial resolution was a longitude/latitude grid of 144 points by
96, respectively (i.e. approximately 2° resolution), and over 88 levels of altitude ranging from the ground to approximately
150 km altitude with approximately 20 levels in the troposphere. Specified dynamics were used, with a nudging towards the
Modern-Era Retrospective analysis for Research and Applications 2 (MERRA2) meteorological data (Randles et al., 2017)
at every time step (30 min) with a weight factor of 0.1 towards the analysis, for temperature and wind fields. Anthropogenic
surface emissions were prescribed for $SO_2$ using the MACCity data set (e.g., Diehl et al., 2012). Carbonyl sulfide (OCS) was
prescribed using data from (Kettle et al., 2002). The simulation presented in this study deals with a multi-annual model ex-
periment starting on January $1^{st}$ 2013 using the CESM1 initial atmosphere state file at that date. The Raikoke and Ulawun
eruptions have been simulated by injecting a volcanic $SO_2$ mass burden into model grid boxes corresponding to the location
of the volcanoes (Raikoke: 48°N and 153°E, Ulawun 5°S and 151°E), over 6 hours, spread evenly between a certain altitude
range for each eruption (see Table 1 for a summary of the model setup) following the method of (Mills et al., 2016). The chosen
$SO_2$ burden of 1.5 Tg is in fairly good agreement with Muser et al. (2020), who calculate 1.37 ±0.07 x $10^9$ kg with TROPOMI
and estimate 1-2 x $10^9$ kg with HIMAWARI data. The model's 2.5° longitude x 1.875° latitude grid resolution means that
the volcanic plumes are initially too dilute in the model compared to reality. This is nevertheless a typical methodology used
in the literature (e.g., Lurton et al., 2018). The timing and altitude injection of the $SO_2$ emissions is based on information





**Table 1.** Characteristics of the model setup accounting for volcanic injections of $SO_2$. The injections have been initialized between 18:00 and 00 UTC.

| Volcano | Date and time | $SO_2$ mass | Injection altitude range |
|---|---|---|---|
| **Raikoke** | 21-22 June 2019 | 1.5 Tg | 9-16 km |
| **Ulawun** | 26 June 2019 | 0. 14 Tg | 16-17 km |
| **Ulawun** | 3 August 2019 | 0. 30 Tg | 17-18 km |

provided by the SSiRC (Stratospheric Sulfur and its Role in Climate) community (SSiRC, 2018) and the results shown in Section 4.1. This SSiRC information relies on $SO_2$ satellite retrievals from IASI (Clarisse et al., 2011), OMI (Ozone Moni-
toring Instrument; Theys et al. (2015)), and MLS/Aura (Krotkov et al., 2008). However, an OMPS aerosol extinction profile shortly after the Raikoke eruption, supports the chosen altitude range of 9-16 km (see supplementary material, Figure A1). The CESM1(WACCM) atmospheric chemistry scheme includes a comprehensive sulfur cycle and key stratospheric nitrogen (NOy), and halogenated and hydrogenated (in particular HOx radicals) compounds. The formation and microphysics of sulfuric acid aerosol particles simulated by the CARMA module are described in detail in English et al. (2011). Following Lurton et al. (2018), aerosol extinctions have been derived at 550 nm and integrated from the tropopause and over the stratosphere to yield a sAOD. In our study, the Raikoke and Ulawun eruptions are simulated by WACCM for a pure sulfate point of view, i.e. ash emissions are not included.

## 3 Raikoke and Ulawun eruptions in 2019

### 3.1 Raikoke

The Raikoke volcano on the Kuril Islands in the Western Pacific Ocean (48.29°N, 153.25°E) is known for its relatively frequent explosive activity (last documented eruptions in 1924 and 1778) (NASA, 2019). Crafford and Venzke (2019) state that a series of paroxysmal eruptions occurred at Raikoke between June $21^{st}$ (18 UTC) and $22^{nd}$ (5:40 UTC) 2019. Some first crude estimations with IASI/Metop-B data indicate $SO_2$ altitudes in the range between 10 and 16 km on June $23^{rd}$ (Aeris, 2018), which is plausible from Himawari data (see below). Hedelt et al. (2019) show plume altitudes ranging from 6-8 km up to 18 km altitude with TROPOMI observations on June $23^{rd}$ and from 11 to 20 km altitude the following day. Sentinel5P/TROPOMI observations indicate an $SO_2$ injected mass of around 1.35 Tg in the Raikoke plume from June $23^{rd}$ (Carn, 2019a). Airplanes flying over the North Pacific had to be redirected (Crafford and Venzke, 2019).

### 3.2 Ulawun

The Ulawun volcano in Papua New Guinea (5°S, 151°E) was identified as one of the 16 'decade volcanoes' by the International Association of Volcanology and Chemistry of the Earth's Interior (IAVCEI) and is therefore known as one of the most potentially destructive volcanoes on Earth (Cas, 2019). Two eruptions occurred during summer 2019, on June $26^{th}$ and August





$3^{rd}$. $SO_2$ injection altitudes between 13 and 17 km are identified with IASI/Metop-B data for the first eruption on June $26^{th}$. For the second and larger eruption, IASI/Metop-B data indicate $SO_2$ altitudes of around 14-17 km for August $3^{rd}$ and $4^{th}$ (Aeris, 2018). For the first eruption Sentinel5P/TROPOMI data suggest a $SO_2$ load of ∼0.14 Tg of the plume, while the second one was a bit larger and data suggest ∼0.2Tg of $SO_2$ (Carn, 2019b). With its tropical location, the eruptions at Ulawun have the potential to have an impact on the lower stratosphere of both hemispheres within the BDC, once injected into the UTLS (Butchart, 2014). Ulawun remained in an active phase with e.g. observed ash plumes in October 2019 up to 3 km altitude (Bennis and Venzke, 2019). By February 2020 only water vapor plumes were observed and the Alert Level remained at Stage 1 (Sennert, 2020).

## 4 Results

### 4.1 Injection and early dispersion of the Raikoke plume

Using a similar method as in Kloss et al. (2020), we attempt an estimation of the injection height using Himawari infrared brightness temperature information, at the moment of the main eruption, and comparing with coincident temperature profiles from ERA5 reanalyses. The brightness temperature of the plume core (not shown) exhibits a plateau at about 225 K within a few hours after the eruption. However, the exact injection altitude could not be identified due to the fact that the temperature profile in the area of the Raikoke is quasi-isothermal in the altitude range between 10 and 24 km (see supplementary material, Fig. A1a). Thus, an univocal attribution of the plume top height at the time of the main eruption is not possible, using this method. There are no CALIOP intersections of the core plume during the early stage. An OMPS aerosol extinction profile, which was observed on 22/06/2019 02:26 at 49°N and 154°E, displays an enhanced aerosol signal at ∼14 km altitude (supplementary material Fig. A1b) that is compatible with previous estimates (e.g., Muser et al., 2020). A sequence of Himawari-8 infrared observations at 20 minute intervals has been used to produce a GIF (GIF in the supplementary material), which displays the complex pattern of plumes of gas and ash emitted by the successive explosions. The exact estimation of the injection altitude is evidently complicated. For the first Ulawun eruption we estimate an injection altitude between 15 and 19 km and for the second between 15 and 18 km, with Himawari brightness temperature and the corresponding ERA5 temperature profile (supplementary material Fig. A2). CALIOP data (not shown) exhibit plumes up to 18 km for the first eruption and 18.5 km for the second eruption. The initial evolution of the Raikoke plume is shown with the Himawari Dust RGB images starting from the 21/06 at 19:00 (Fig. 1). The Dust RGB product is used, instead of the Ash RGB product, because it is more sensitive for large satellite viewing angles, which is the case for the region of interest for Raikoke. This plume is initially composed of ash (reddish colors, in Fig. 1), with also some evidence of $SO_2$ (yellowish colors, in Fig. 1). The remaining greenish and pinkish colors indicate the presence of water clouds around the volcanic plume. Over June $22^{nd}$, the plume disperses eastward of the volcano and separates into an ash dominated component in the south and a $SO_2$ dominated component in the North (see the two upper rows of Fig. 1). In the following days, the ash plume is rapidly diluted or sediments, and cannot be further followed. The $SO_2$ plume instead persists and, from June $23^{rd}$, stops mowing eastward to wrap upon itself and get trapped for several days within the cyclonic circulation of the Aleutian low which was unusually strong for this summer period. As a





consequence, the confined plume remains compact and exhibits a number of dense patches and filaments that are well defined
      in the Himawari images, reaching locations as far as Alaska and central Russia, as visible from IASI $D_{SO2}$ observations (Fig.
      A3). CALIOP sections of these patches on June $25^{th}$ and $26^{th}$ (not shown) exhibit aerosol plumes up to 15.5 km. We found
      no confirmation of the rise to 22 km within a few days reported in the modelling study of (Muser et al., 2020). After June
      $25^{th}$, the SO$_2$ plume gets more diluted and is possibly converted to sulfate aerosols. The presence of a compact SO$_2$ plume,
after ash removal, is supported by the strong detection of SO$_2$, i.e. $D_{SO2}$ significantly larger than 1.0, that is obtained with the
      high-spectral-resolution observations of IASI, starting from 23/06/2019, at about 9:00 am (morning overpass, Fig. 2b). The
      intensity of the $D_{SO2}$ detection parameter decreases in the following days (Fig. 2c,d), as the plume dilutes and possibly a part
      of the SO$_2$ converts to sulfate aerosols.

### 4.2   The global dispersion of the Raikoke and Ulawun plumes with OMPS observations and WACCM simulations

After the first atmospheric processing following the injection in the UTLS, including the entrainment into the storm discussed
      in Sect. 4.1, the Raikoke plume entered the global, lower stratosphere. To study the global, stratospheric distribution of the
      enhanced aerosol layer during the year following the Raikoke eruption in June 2019, we use the OMPS-LP aerosol extinction
      observations for one year following the eruption (Fig. 3 and 4a) combined with WACCM simulations (Fig. 4b and c). We
      investigate the possible interaction of the Raikoke and Ulawun perturbations on the stratospheric aerosol layer properties and
their impacts on the radiative balance.

      The initial injection and early plume dispersion during the first week, seen in Figs. 1 and 2 with Himawari and IASI observa-
      tions, is not as evident looking at the global view from OMPS (Fig. 3a). A blind stage like this for observational data, was also
      found after the Sarychev eruption (Haywood et al., 2010). Fig. 3b shows a first clear enhancement north of the Raikoke loca-
      tion spreading towards the east during the first week of July (more than one week after the initial injection). This enhancement
of the aerosol extinction is most likely due to the conversion of the SO$_2$ plume to sulfate aerosols (i.e. longitude dispersion
      occurred faster than the conversion to H$_2$SO$_4$). Enhanced sAOD values in Fig. 3b further west (i.e. above Europe) can be
      attributed to the plume from the Alberta fires in Canada from June 2019. During the following weeks and months the sAOD
      increases throughout all longitudes north of the Raikoke location. In August (Fig. 3d) the AOD is increased by around a factor
      of 5 compared to prior Raikoke conditions in Fig. 3a, reaching values larger than 0.025 for the integrated sAOD (calculated
from tropopause upwards). Starting from August an increase in AOD is also evident south of the Raikoke location. Even in the
      mid-latitudes and tropics an enhanced signal is visible in Fig. 3d and e. This might result from an efficient transport within the
      horizontal circulation of the Asian monsoon anticyclone (AMA). This is supported by the fact that no aerosol enhancement is
      visible within the AMA core (Fig. 3c and d). A mixing from the aerosol plume from the second Ulawun eruption (August $3^{rd}$)
      is possible as well. From July to October (Fig. 3c-f) the transport barrier of the AMA leads to a low bias of sAOD values (i.e.
air masses with increased aerosol do not pass into the area of the AMA). From September 2019 to May 2020 AOD values sys-
      tematically decrease as the downwelling of the lower branch of the BDC in the NH (northern hemisphere) intensifies and due
      to wet/dry deposition once aerosols are back in the troposphere. However, values remain elevated compared to prior Raikoke
      conditions even one year after the eruption (Fig. 3i).



**Figure 1.** Himawari Dust RGB images from 21/06/2019 to 28/06/2019, over the region of Raikoke. Red: ash; Pink to violet: dust; Yellow: mixture of ash and $SO_2$, Green: thick and thin mid-level clouds or cirrus clouds. The contour lines are plotted for the Montgomery potential on the potential temperature surface 340 K and indicate the mean atmospheric circulation. The image frame is expanded from the first panel to the last to follow the dispersion of the plume.



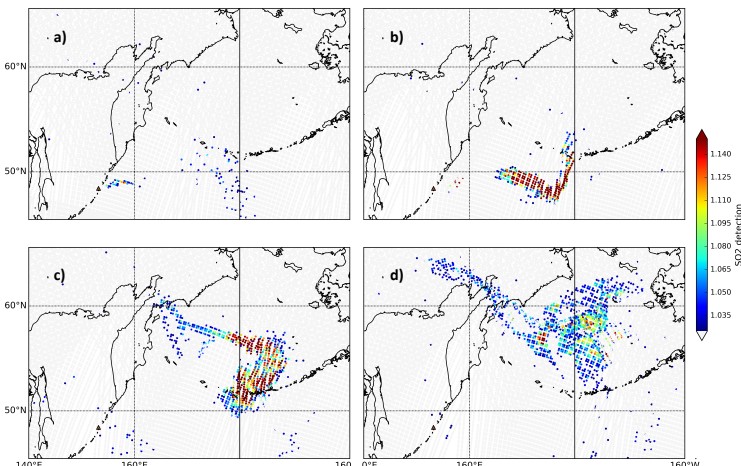

**Figure 2.** IASI SO$_2$ detections D$_{SO2}$, for the morning overpasses (about 9:00 LT) (a) for June $22^{nd}$, (b) $23^{rd}$, (c) $24^{th}$ and (d) $25^{th}$ 2019.

Besides Raikoke, OMPS detects elevated aerosol extinction values during the days following both Ulawun eruptions. Figure 3b
and d show enhanced sAOD values during the two weeks following each eruption. The second Ulawun eruption gives a higher
AOD signal in terms of spatial extent and maximum value (by a factor of around 2, Fig. 3b and d). The aerosol plume from the
first Ulawun eruption (June $23^{rd}$) is mostly propagating eastwards at the equator (second panel in Fig. 3b and c). The plume
from the second eruption was distributed in both directions in the tropics (east and west). The eastward transport dominates,
which depends on the vertical distribution of the aerosol and the phase of the QBO (quasi-biennial oscillation). During October
and November (Fig. 3 f and g) the tropical stratosphere is enhanced with increased aerosol extinction values. We estimate
a circling of the Earth in the tropics of the second Ulawun eruption in the vicinity of one month. The tropical background
aerosol 1 month after the Ulawun eruptions is increased by a factor of around 3, reaching sAOD values as high as 0.02, in a
very limited latitude range. In May 2020 the AOD signal in the tropics remains enhanced. We attribute enhanced sAOD values
from August 2019 onwards south of 30°S, which are clearly separated from the increased values in the tropics, to a horizontal
tropopause crossing, originating from the Ulawun eruptions (further discussed below). During the end of 2019/ beginning of
2020 historically severe wildfires occurred in Australia. Through the formation of pyro-convection a significant part of smoke
particles was injected in the stratosphere (Khaykin et al., 2020). Most of the enhanced AOD in the SH in Fig. 3h and i originates
from those fires, likely mixing with the remaining enhanced aerosol signature from the Ulawun eruptions. While in this study,
we focus on the complex situation of the global transport of the Raikoke and Ulawun aerosol plumes, in the tropics and NH,
the global impact of the Australian fires mostly impact the lower stratosphere in the SH (southern hemisphere). They pose a
unique example of stratospheric perturbation from an extreme wild fire event and should be investigated in a separate study.

Another representation for the horizontal distribution and evolution of sAOD (latitude/time Hovmöller plots, averaged over
all longitudes) is presented in Figure 4a for OMPS observations and Figure 4b and c for the WACCM simulations. While
OMPS observations show a clear increase of AOD only around 1 month after the eruption north of the Raikoke position



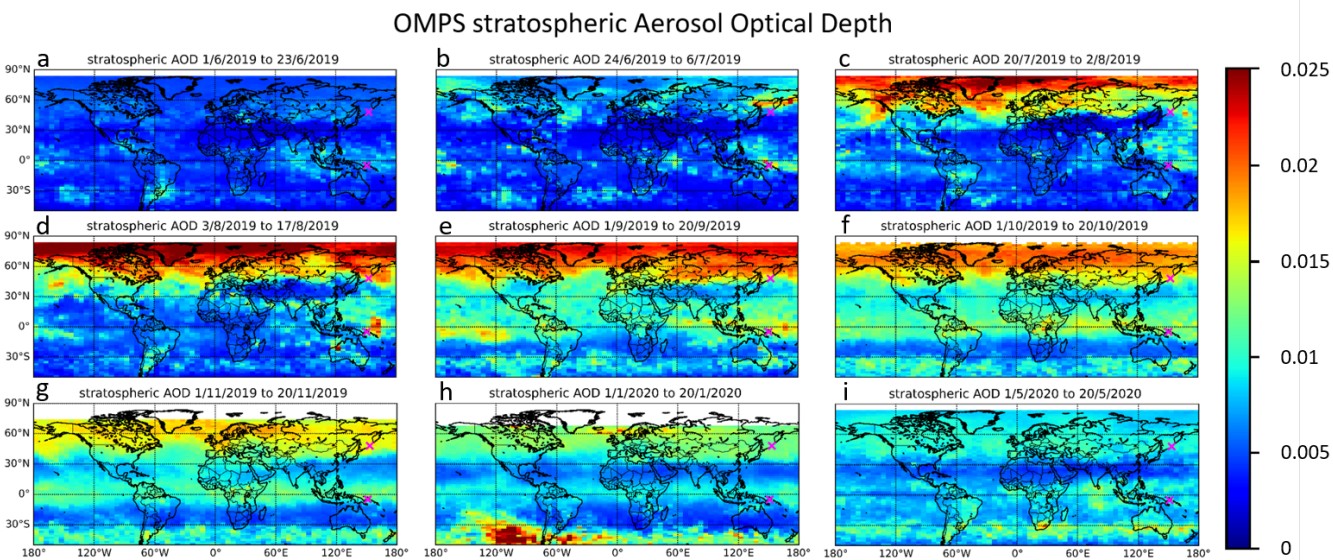

**Figure 3.** Global OMPS sAOD averaged over the indicated time frames (a-i: June 2019 to May 2020). The locations of Raikoke and Ulawun are indicated with magenta crosses. White areas in the North are not covered by OMPS measurements.

(48°N), WACCM already gives a strong signal during the initial injection (Fig. 4b). During the first few months (until October) WACCM sAOD values are significantly larger. These deviations may be due to different reasons. In part, this depends on the different wavelengths of OMPS observations and WACCM outputs: 550 nm for WACCM and 676 nm for OMPS. The lack of ash emissions in the WACCM simulations may also cause different dynamics of sulfate aerosols formation. This is a well known effect in volcanic eruption modelling and similar deviations between limb observations and modelling of sulfate aerosol

plumes build-up have been observed e.g. for the Sarychev eruption (Haywood et al., 2010, see their Fig. 5). Furthermore, the AOD values from the model simulation seem to decrease faster (Oct.-Dec. 2019) than for OMPS. For both Ulawun eruptions (June 26$^{th}$ and August 3$^{rd}$), OMPS data show some AOD perturbations after the first eruption and more significantly elevated values after the second eruption. Like for the Raikoke eruption, WACCM shows immediate and stronger signals during the weeks following the eruptions, but decreasing faster. While for OMPS observations a significant impact (sAOD around 0.01)

of the second Ulawun eruption is still apparent in the tropical stratosphere by the end of the year 2019, in the model comparable values are found in October and by the end of 2019 the sAOD has values down to 10 times smaller than for OMPS. The model shows a faster decrease. Thanks to the modelling capabilities, we have isolated the impact of the Ulawun plumes (Fig. 4c), so to analyze the possible crossed-impact of the eruptions from this volcano in the northern regions already affected by the Raikoke eruption and, vice versa, to detect possible impacts of the Raikoke plume in the tropics and SH. As described for Fig.

3 the AOD enhancement starting from July 2019 at 40°S is clearly separated from the Ulawun impact on the tropical strato-sphere and can result from a horizontal tropopause crossing of the aerosol plume towards the south (Fig. 4a). This hypothesis is confirmed by the model simulation in Fig. 4b and c, where only volcanic sources of stratospheric aerosols are considered.





However, it has to be noted that WACCM simulations reveal elevated sAOD values in the SH originating from the Raikoke eruption (see also Fig. A4 in the supplementary material). Such a feature is not confirmed with CLaMS passive air mass tracer
simulations (not shown). For an accurately defined altitude level of tropopause crossing more analysis would be needed, which goes beyond the scope of this study. A similar enhancement due to tropopause crossing in the North from the Ulawun plume would possibly interfere with the interpretation of the global distribution of the Raikoke plume. However, the Ulawun-only simulation of Fig. 4c indicates that very limited to no transport of the Ulawun plume occurred to the North via an horizontal tropopause crossing. Because the influence of the Ulawun eruption on the SH seen by OMPS is well reproduced by the model,
we trust this conclusion. However, a transport during the winter months (Nov/Dec/Jan) also towards the North within the BDC, as seen following the Ambae eruption in 2018 (Kloss et al., 2020), is likely. Even though such a feature is not clearly visible in OMPS observations (Fig. 3 and Fig. 4a), we believe that an already enhanced aerosol layer in the North (following the Raikoke eruption) masks this transport towards the North in the winter months. By the end of the year, WACCM simulations in Fig. 4b and c show low sAOD values, which is why the model data potentially miss this feature as well. Consequently, we cannot
rule out that Ulawun air masses have interfered with the Raikoke evolution. Figure 4b shows higher sAOD values in the tropics and SH compared to Fig. 4c. Hence, the Raikoke eruption had a significant impact on the tropical stratosphere. The sAOD for the respective Raikoke WACCM simulation is presented in the supplementary material (Fig. A4). As also seen in Fig 3h, the enhanced aerosol signature starting from the end of 2019 in the SH is attributed to the aerosol plume of the Australian wildfires 2019-2020.

Discrepancies between the model output and OMPS observations are expected, especially following the Raikoke eruption, because of the following reasons. WACCM does not account for ash particles. In a recent study by Muser et al. (2020) a burden of 0.4-1.8x$10^9$ kg is estimated for ash particles (with a diameter <32 $\mu$m). Whether ash is included or not determines the chemical evolution, dynamics and aerosol load. The WACCM simulations can therefore only be seen as a pure sulfate point of view with the mentioned limitations. Furthermore, the determination of the altitude range of the plume injection is very challenging.
The injection altitude in WACCM is based on IASI observations. However, as discussed in Sect. 3, the exact determination of the injection altitude is impossible, at least for the specific atmospheric conditions during the Raikoke eruption. The plume dispersion (Lachatre et al., 2020) and its chemical/microphysical evolution depends strongly on the initial injection altitude. Any information about the $SO_2$ injection altitude cannot be derived from ash because different altitude levels can be reported for $SO_2$ and ash (Vernier et al., 2016). The same goes for timing and burden of the plume injection. Here, we assume an evenly
distributed injection (vertically and time-wise), which is a schematic estimate, but for sure causes discrepancies compared to observations and reality. The sulfate burden injected was taken from the SSiRC community based on the IASI data set, which agrees well with estimations from Muser et al. (2020) with 1.37 ±0.07 x $10^9$ kg from TROPOMI and 1-2 x $10^9$ kg for Himawari-8. It can be assumed that different instruments and models with different set ups will come up with varying values for the burden (as seen after the Sarychev eruption e.g. Günther et al. (2018); Kristiansen et al. (2010); Krotkov et al. (2010)).
The issue of different models and instruments leading to different scientific conclusions is addressed in Fromm et al. (2014). Other aerosol sources (e.g. from other, minor volcanic eruptions or dust) are not included in the model. For OMPS, we use the full, non filtered data set of aerosol extinction values. Hence, potential cloud signatures are included in the observations,



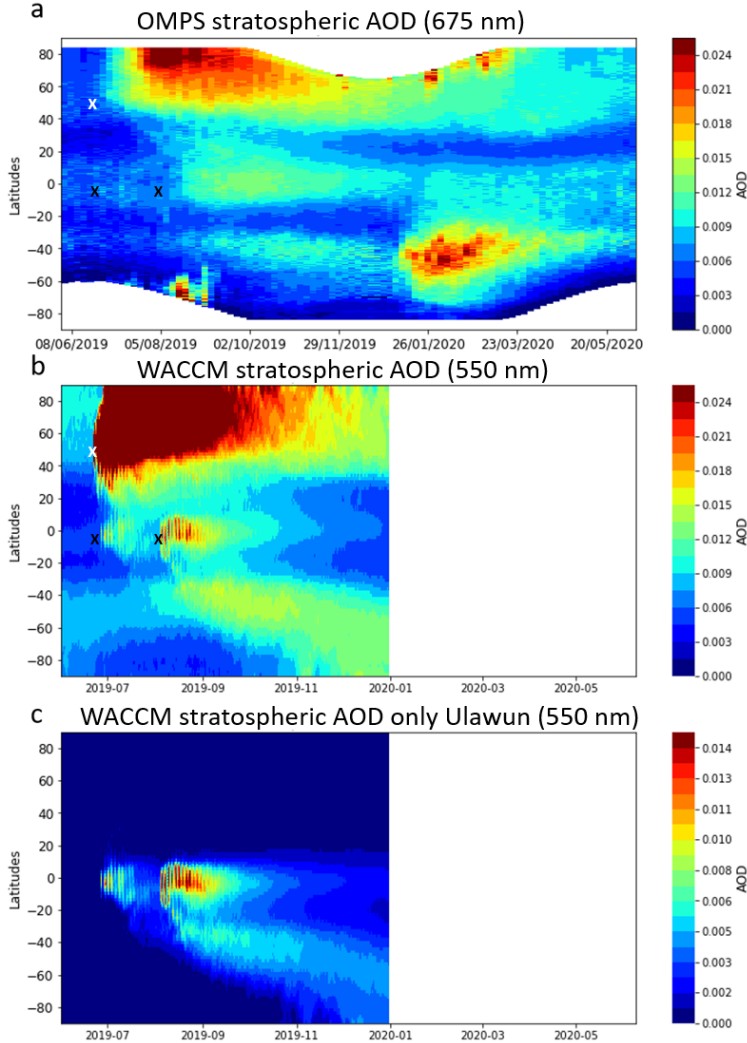

**Figure 4.** Latitude/time global distribution of the longitude-averaged stratospheric AOD. (a) for OMPS observations and (b) the integrated stratospheric column for the WACCM simulation for both eruptions and (c) for the Ulawun-only WACCM simulations. Crosses symbolize the position and timing of the eruptions, white for Raikoke and black for Ulawun.

but excluded for the WACCM simulation. The sampling of any observational instrument compared to that of a global model is not comparable. The WACCM global simulations are performed twice a day while OMPS reaches a global coverage every ~3 days. The sampling bias could be prevented by sampling WACCM data according to the OMPS orbit. However, while this bias can mean that very local features are missed by OMPS, it cannot explain time delays in the order of weeks. Despite these marked differences, the general dynamics of sulfate aerosol formation/removal, from observations and simulation, is rather consistent in terms of the impacted latitude bands.



### 4.3 The global distribution of the Raikoke and Ulawun plumes with a passive air mass tracer

A complementary overview of the dynamics of both volcanic plumes, once injected into the UTLS is given using an air mass origin tracer with CLaMS (Fig. 5). We choose two initialization boxes in space and time corresponding to the rough respective injection locationsof the volcanic plumes. For the Raikoke eruption the initialization box is chosen according to observations by Hedelt et al. (2019) (at 11-18 km, 335-460 K, from 163°E to 170°W and 49-62°N for the $23^{rd}$-$24^{th}$ of June 2019). This is equivalent to the position of the plume during the storm entrainment (see Figs. 1 and 2). For simplicity, we ignore the

minor impact of the first Ulawun eruption (from June $26^{th}$). For the larger eruption at Ulawun (August $3^{rd}$), we define a rough injection box from 137-178°E, 10°S-5°N from 14 to 17 km altitude, 350-385 K, on August $3^{rd}$ and $4^{th}$ of 2019. After initialization, the tracer is advected passively during the subsequent months. This simulation is a simple way of illustrating the plume's global transport in the UTLS throughout the weeks following the respective eruptions, integrated over all altitudes. The simulation cannot be taken for quantitative estimations for the following reasons. First, the chosen initialization is given

in a box shape, whereas the real injection does not appear in the shape of a box. Therefore, many trajectories in this simulation do not necessarily correspond to an actual plume air parcel during injection. Second, in this simulation we use a passive tracer, with no chemical/microphysical processes being taken into account. Finally, the injected burden and related quantitative factors are not accounted for in the CLaMS simulations, as the Raikoke and Ulawun air mass tracers are equally represented. However, as CLaMS transport is driven by the newest reanalysis (ERA5) the simulation provides a reliable diagnostic of the air mass

transport from the volcano region (initialization box).

Once initialized after the Raikoke eruption, the air mass tracer is transported towards the East, which is consistent with OMPS observation (see Fig. 3). By mid-July (roughly within 3 weeks after the eruption), the plume tracer has circled the Earth on latitudes mostly north of the Raikoke location. At the beginning of July the main bulk of the air mass tracer remains west of the Atlantic Ocean. Therefore, the sAOD enhancement above Europe observed by OMPS in Fig. 3b does not originate

from Raikoke, but rather from forest fires in Alberta, Canada. The plume air mass transport is largely consistent with OMPS observations, as by the end of July (Fig. 3) enhanced AOD values are apparent throughout all longitudes, mostly north of the Raikoke position. For the CLaMS simulation a clear signal of the tracer is visible around the area of the AMA from end-July until mid-September, which is also consistent with OMPS data (Fig. 3c-e). By mid-August a small percentage of the initialized Raikoke tracer has reached the tropics in the CLaMS simulations. Such a transport can also be seen from OMPS and WACCM

data in Fig. 4a and b in July/August 2019 (with sAOD values below 0.01 for OMPS). As seen for OMPS data, the plume tracer initialized according to the second Ulawun eruption is transported east- and westwards, with a dominating component towards the East. The CLaMS air mass tracer suggests a circling of the Earth in the tropics within less than one month (which agrees with OMPS data as well, c.f. Section 4.2). Already during the first month after the Ulawun eruption in August, the simulation with CLaMS suggests that an overlapping of air mass tracers for the Raikoke and Ulawun eruptions is possible in the tropics.

Starting from September the air mass tracer for the Ulawun eruption remains largely in the tropics (between 0-30°S), slowly expanding towards the North and South.

Even though CLaMS simulations neither take any chemical/microphysical processes into account nor possible lifting due to





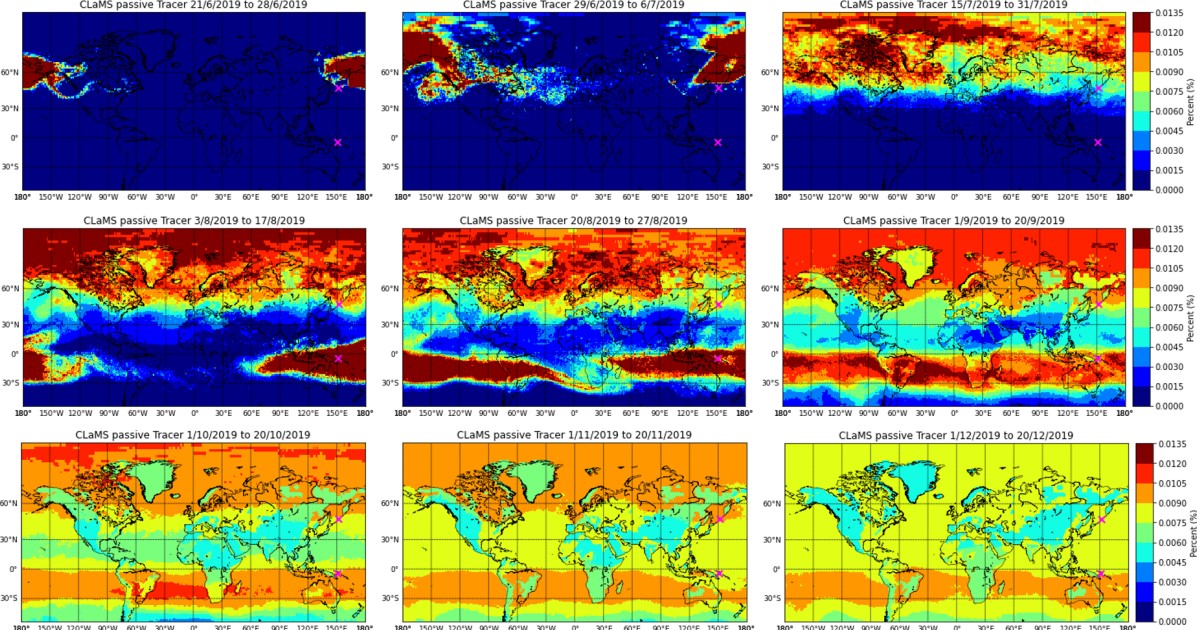

**Figure 5.** Mean column air mass fraction from the Raikoke and Ulawun plumes, calculated from integrating the passive CLaMS air mass tracers over all vertical levels. The two main eruptions (Raikoke June $21^{st}$ and Ulawun August $3^{rd}$) are equally included. Initialized boxes are selected according to Himawari and IASI observations. The Raikoke eruption is initialized from June $23^{rd}$ -$24^{th}$, 163°E-170°W, 49-62°N and 335-460 K. The Ulawun eruption is initialized from $3^{rd}$ to $4^{th}$ of August, 137-178°E, 10°S-5°N and 350-385K. The magenta cross symbols indicate the location of the two volcanoes.

aerosol-radiation-dynamics (suggested to play a crucial role for the Raikoke eruption in Muser et al. (2020)), the horizontal distribution for the CLaMS simulation is well represented in terms of dynamics.


## 4.4 Vertical distribution

Figure 6 shows the vertical distribution of aerosol extinction values, and its evolution, around the location of the volcano. The initial injection phase after the Raikoke eruption is more evident for the WACCM simulation than for OMPS observations (Fig. 6a and b). In the model, the aerosol plume rises from around 15-20 km altitude during the month following the eruption,

while OMPS shows maximum altitude values of the aerosol plume rise with a slower rate from around 15 to 22 km altitude (1.5 km per month). The approximate descending rate in OMPS data, from November 2019 to February 2020, of around 2 km per month reflects a contribution from both sedimentation processes and the descending branch of the BDC. Increasing aerosol extinction values in spring 2020 around the tropopause are a recurrent seasonal feature, independent from Raikoke perturbation.

For the August eruption of Ulawun, both WACCM and OMPS show a plume rising up to 19 km (first eruption) and 20 km





(second eruption), directly after the respective eruptions (Fig. 6c and d). A subsequent transport to ∼21 km in the area around the volcano is also shown in observations and reproduced in the model. One month after the eruption, the signal of the dispersed plume is at higher altitudes in the observations than in the model. This can potentially reflect an underestimation of the amount of $SO_2$ initially injected in the model. As seen in Fig. 4, OMPS reveals increased aerosol extinction values even 10 months after

the second Ulawun eruption, while WACCM values seem almost back at background conditions within 5 months. The large differences between OMPS observations and the WACCM simulation seen in the troposphere can be explained by clouds and other tropospheric sources of aerosols, which are not included in the model. We focus on the transport in the lower stratosphere, rather than the troposphere, therefore, those differences are of no interest in this study.

The panel series in Fig. 6e shows, in a similar manner to what is shown in Chouza et al. (2020) (in their Figure 7, using

CALIOP data), the vertical distribution of mean aerosol extinction OMPS values averaged over all longitudes for each month from June to December 2019. Following the Raikoke eruption, a clear enhanced aerosol extinction signal is visible north of the Raikoke location (48°N), rising from ∼16 km to 17.5 km from July to August (∼1.5 km per month: ∼0.3 mm/sec). A clear rise up to altitudes at around 25/26 km from the Raikoke plume as discussed in Chouza et al. (2020) is not apparent in Fig. 6e. Slightly enhanced aerosol extinction values following the Ulawun eruption appear in the tropics in August at above 17 km.

The Ulawun plume remains largely in the tropics and rises within the ascending branch of the BDC (∼1 km per month: ∼0.4 mm/sec from September to December).

## 4.5   In the context of other recent events

Figure 7 shows mean sAOD estimations for OMPS, SAGE III/ISS and in situ LOAC observations from France. The mean

sAOD from the OMPS and SAGE III/ISS aerosol extinction observations are at 675 and 676 nm, respectively. The dense sampling, reaching high latitudes from OMPS gives confidence in the representation of the overall AOD evolution (Fig. 7a). While we present 3-day averages for the OMPS data set, we calculate 30-day averages for SAGE III/ISS, to account for the much sparser sampling of SAGE III/ISS.

The timing and total value of sAOD enhancements for OMPS and SAGE III/ISS (Fig. 7a and b) following the Canadian

wildfires in 2017, the Ambae eruptions in 2018 and the Raikoke/Ulawun eruption in the different latitude bands agree very well. Observed peak sAOD values by SAGE III/ISS are by ∼10% higher than OMPS values for most latitude bands, which is consistent with the difference of ±10% found by Chen et al. (2019) following the Ambae eruption. Peak values in the 30 - 50°N latitude band are significantly higher following the Raikoke eruption for SAGE III/ISS values, which is likely due to the sparse sampling. Compared to the sAOD impact of the Canadian fires in 2017, the Raikoke eruption led to 2.5 times higher

AOD peak value north of 50°N (for OMPS and SAGE III/ISS data in Fig 7a and b). Particular sAOD enhancements from the two stratospheric fire events in 2019 (Alberta in June and Siberia in July) are not visible. The Raikoke plume has likely mixed with the plumes of the fire events, however, compared to Raikoke the fire signature is small. The impact of the Ulawun eruption on the tropical sAOD from OMPS is by a factor of around 1.5 higher than what was observed for Ambae (factor of 1.8 with SAGE III/ISS data). For the past three years, including the impact of the Canadian fires, Ambae eruption and Australian fires





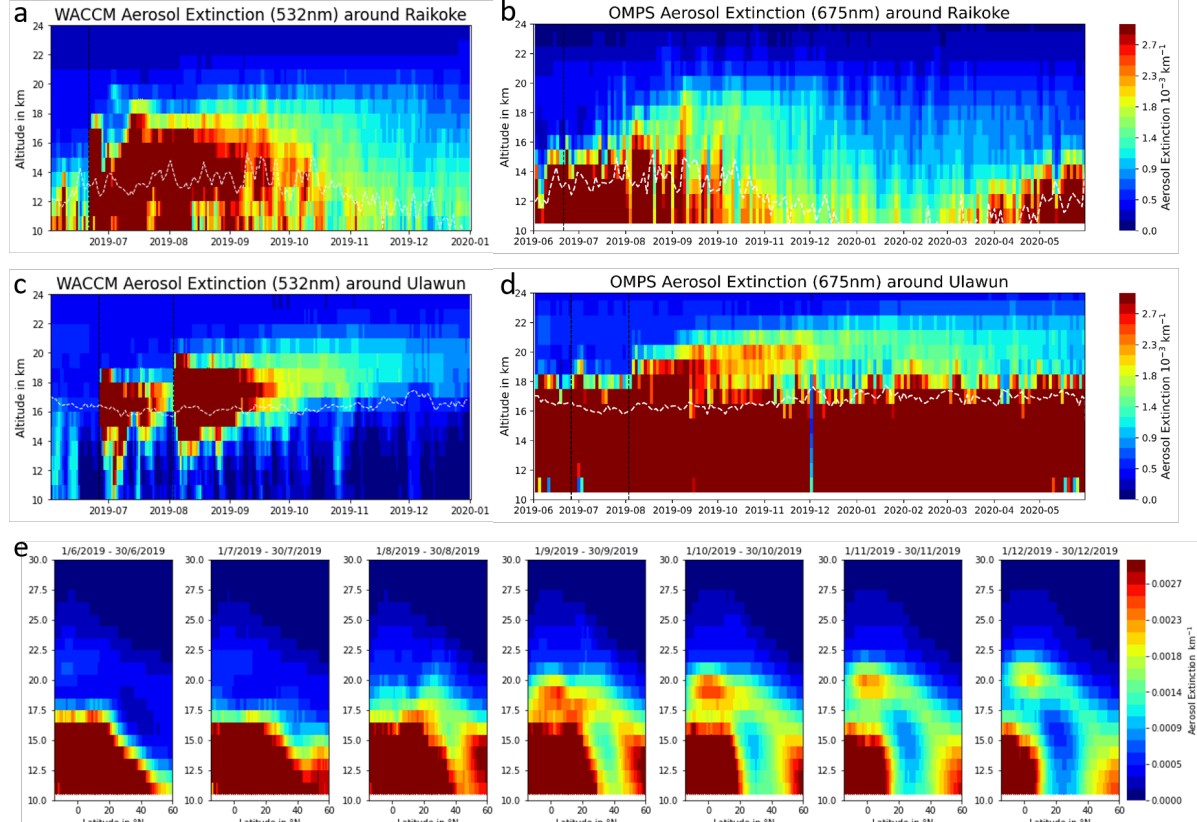

**Figure 6.** Vertical Aerosol extinction distribution at the location of the volcanoes. (a) WACCM simulation around the Raikoke location (30 - 55°N and 145 - 175°E) and (b) OMPS observations respectively. (c) and (d) respectively for the Ulawun eruption (0 - 10°S and 145 - 175°E). White dashed lines represent the averaged tropopause altitude. The timing of the Raikoke and both Ulawun eruptions is indicated by the black, dashed lines. (e) OMPS aerosol extinction monthly averages over all longitudes from June to December 2019.

on the sAOD, the Raikoke and Ulawun eruptions have had the highest impact on NH sAOD levels. Even one year after the eruptions, AOD values on latitudes higher than 50°N are elevated and comparable to sAOD values from the peak phases of the Canadian fire event. Other similar comparisons one year after the Raikoke and Ulawun eruption are not possible, because of the impact of the recent Australian wildfires (AOD increases seen from end of 2019 onwards). OMPS and SAGE III/ISS data suggest a comparable but smaller sAOD impact for the Australian fires than for the Raikoke eruption (30-50°S compared to 50-90°N).

LOAC in situ observations in central France (Figure 7c) show a maximum AOD value in August, which coincides with the satellite observations in Figure 7a and b. Furthermore, this is consistent with Figure 3, which also shows enhanced sAOD values above France in August 2019. For LOAC, only partial AOD (in terms of particles size) have been derived for LOAC in situ data, i.e. in the range from 0.2-0.7 $\mu$m, to avoid spurious aerosol extinction enhancements resulting from the presence





of low-concentrated micrometer-sized particles, for instance coming from the balloon flight chain above the instrument or corresponding to the "background" meteoritic population (e.g., Renard et al., 2008; Murphy et al., 2014). As a result, the LOAC AOD values cannot be directly compared with OMPS. The in situ AODs reveal a significant enhancement over the 2019 summer-autumn period above France. Following the Raikoke eruption, the in situ data present an oscillating behaviour with some low values in late 2019 (especially the October measurement in Fig. 7c). This could reflect the sparse and very local

sampling of in situ observations and could also be explained by a still inhomogeneous volcanic plume at this period. The slight increase in the observed AOD in April 2019 can be related to remnants of the midlatitude signature of the Ambae eruption (Kloss et al., 2020) and could reflect that background aerosol conditions were not reached in the stratosphere for the period before the Raikoke eruption, which is consistent with OMPS and SAGE III/ISS observations in Fig. 7a and b.

## 5 Optical properties and the global impact on the radiative balance

The multispectral SAGEIII/ISS observations are used to further characterize the optical properties of the Raikoke and Ulawun plumes and to estimate their radiative forcing (RF). Despite their sparser spatiotemporal sampling, with respect to OMPS, the solar occultation geometry of SAGE III/ISS observations is associated with a better signal-to-noise ratio. Figure 8a,b show the average Raikoke- (panel a) and Ulawun-attributed (panel b) SAGE III/ISS sAOD, at the different available wavelengths be-

tween about 449 and 1020 nm. The Raikoke-perturbed spatiotemporal interval has been considered as the longitude-integrated latitude bands between 40 and 70°N, in the period from the eruption to end of September 2019. While at periods later than September 2019 the stratosphere is expected to be still somewhat perturbed by late Raikoke plume, the selected period is chosen to be representative for both peak and declining volcanic perturbation (see Fig. 4a). To get a more detailed characterization of the plume and its impact, we subdivided the overall latitude range chosen for Raikoke into two sub-intervals: 40-55°N

and 55-70°N. It is important to mention that latitudes higher than 70°N are very sparsely sampled with the SAGEIII/ISS orbit. Furthermore, higher impacted regions in terms of stratospheric aerosol are possibly partly missed by SAGE III/ISS. The Ulawun-perturbed spatiotemporal interval has been considered as the longitude-integrated latitude bands between 20°S and 15°N, in the period from the eruption to end of November 2019, which encompasses the whole evolution of the Ulawun plume. For both eruptions, a corresponding background atmosphere has been chosen, in a clear period at as much as possible

similar seasonal conditions, as a baseline for both the sAOD and the RF estimations: September $1^{st}$ to $15^{th}$ 2018, at both 40-55°N and 55-70°N, for Raikoke, and June $15^{th}$ to $30^{th}$ 2018, at 20°S-15°N, for Ulawun. The respective background is subtracted from both Raikoke- and Ulawun-attributed sAOD values, to obtain plume-isolated sAODs for both eruptions. For Raikoke, the whole average sAOD (plume plus background) reaches values as large as 0.045 (at 449 nm) to 0.030 (at 1020 nm), at 55-70°N and 0.030 to 0.020, at 40-55°N. The impact of Raikoke is significantly larger at higher latitudes. The plume-isolated

Raikoke sAOD, i.e. with the background subtracted, reaches values as large as 0.035 to 0.025 (55-70°N) and 0.020 to 0.015 (40-55°N), depending on the wavelength. Comparing the sAOD at 550 nm of Andersson et al. (2015), for the past moderate eruptions of Sarychev, Kasatochi and Nabro (∼0.012, 0.012 and 0.09), with our estimations for Raikoke, this latter eruption

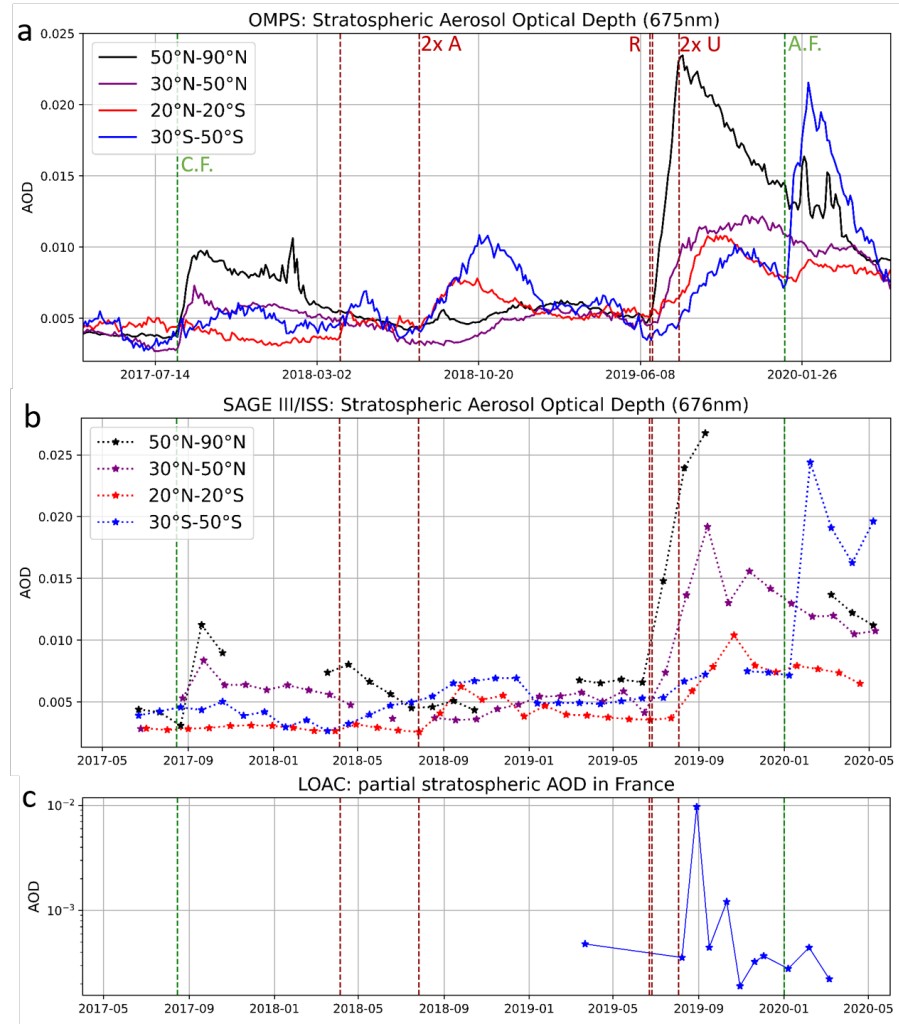

**Figure 7.** (a) 3- day mean sAOD from OMPS aerosol extinction values (from tropopause altitude up to 30 km) averaged over four latitude ranges (50-90°N, 30-50°N, 20°S-20°N and 30-50°S). Vertical lines represent the event dates of the Canadian Fires (C.F.), both Ambae eruptions (2xA), Raikoke eruption (R), both Ulawun eruptions (2xU) and the Australian Fires (A.F.). (b) Same as a, but with SAGE III/ISS measurements for 30-day averages. The maximum value is an average over the month of September (30/8/2019-29/9/2019) with 221 measurement profiles in a latitude range from 50-61°N (black line in b). The respective averaged AOD for August (31/7/2019-30/8/2019) considers 51 profiles from 50-58°N. (c) Derived partial sAODs for balloon borne LOAC aerosol concentration observations from Ury in France, for particle sizes from 0.2-0.7 $\mu$m up to 23 km altitude.

displays sAOD perturbations at least twice as large as past eruptions. Ulawun exhibits significantly smaller plume-isolated sAODs: from 0.010 (at 449 nm) to 0.0025 (at 1020 nm), hence, comparable with the Ambae eruption in 2018 (Kloss et al., 2020) and smaller than Sarychev, Kasatochi and Nabro (Andersson et al., 2015). It is interesting to notice how the spectral





variability of the plume-isolated sAODs, while clearly decreasing with the wavelength, as expected, is somewhat more steep for Ulawun than Raikoke. This could suggest a more homogeneous small-sized sulfate aerosol composition of the Ulawun plume and the possible presence of either some ash or carbonaceous or larger sulfate-coated ash or carbonaceous particles in the Raikoke plume. Bulk estimations of the Angström exponent (AE) of the background and volcanically perturbed aerosol layers, for both volcanic eruptions, can be determined exploiting the spectral variability of the sAOD. For both Raikoke and Ulawun, a pristine average AE of about 1.7 is estimated using the background sAODs. While the Ulawun eruption did not significantly perturb the average AE (AE of the Ulawun-perturbed stratospheric aerosol layer of about 1.7), the Raikoke eruption modified significantly this parameter (AE of the Raikoke-perturbed stratospheric aerosol layer of about 1.2). The AE is an optical proxy of the mean particle size in an aerosol population, with larger AE values associated with smaller particles, and vice-versa. While values approaching 2.0 are typical of smaller sulfate aerosols-dominated aerosol populations, values of 1.2 can be associated with significantly larger particles. Thus, Raikoke perturbed the stratospheric aerosol layer by producing significantly larger particles than the background. We calculated the shortwave RF of the Raikoke and Ulawun plumes using the UVSPEC radiative transfer model (see Sect. 2.7 for the setup of the model and calculations). As input parameters for the model, the SAGE III/ISS volcano-attributed aerosol extinction profiles discussed above are used. While these are measured parameters, some assumptions must be done on two non-measured optical properties of the plume: the single scattering albedo (SSA, an optical proxy of the absorption properties of the plume) and the phase function, summarised by the scalar asymmetry coefficient (g, a metric of the forward/backward scattering properties, linked to the size and composition of the particles in the plume). In the past, very weakly absorbing plumes, composed of small particles, have been proposed for volcanic perturbations of the upper-tropospheric and stratospheric aerosol layer (e.g., Sellitto et al., 2017; Kloss et al., 2020), based on the hypothesis that these are mainly composed of tiny secondary sulfate aerosols. In our case, both parameters are very uncertain and, as discussed above, the presence of larger ash-coated or ash particles cannot be excluded. For this reason, we performed several RF estimations with a range of SSA (from 1.00, typical of non-absorbing particles, down to 0.97, thus partly absorbing particles) and g values (from 0.50, typical of very small particles, up to 0.85, linked to significantly larger particles). The regional RF estimations, in the latitude bands 40-55°N and 55-70°N (Raikoke) and 20°S-15°N (Ulawun), are shown in Fig. 8c,d, for the different values of SSA and g assumptions. By scaling the SAGE III/ISS extinction with the OMPS-derived AOD ratio 55-70°N/70-90°N, the RF has been extrapolated to 70-90°N and is also shown in Fig. 8c,d. Regional RF values as large as -2 to -3 W/m$^2$ are found for Raikoke, at both TOA and surface, in the 40-55°N and 55-70°N, respectively, for the assumption of very small (g=0.5) and very reflective (SSA=1.0) particles, which is linked to a significant cooling of the regional climate system and a very limited, if any, energy absorbed by the plume. The TOA RF at the highest northern latitudes (70-90°N) is found to values as large as -5 W/m$^2$ but this estimation has to be taken with caution (as discussed above, it is based on an extrapolation). For smaller SSA, the TOA and surface RF start to deviate significantly (larger surface than TOA RF), thus indicating a significant absorption of radiative energy into the plume. This energy imbalance and the possible resulting radiative heating of the plume can be a possible reason for the observed lifting, shown in Fig. 6e; this hypothesis requires further investigation. The assumption on the asymmetry parameter g dominates the uncertainty of the RF estimations (compare the error bars of Fig. 8c and d). It is important to mention that all these RF estimations are based on the assumption of clear-sky, so these are just a



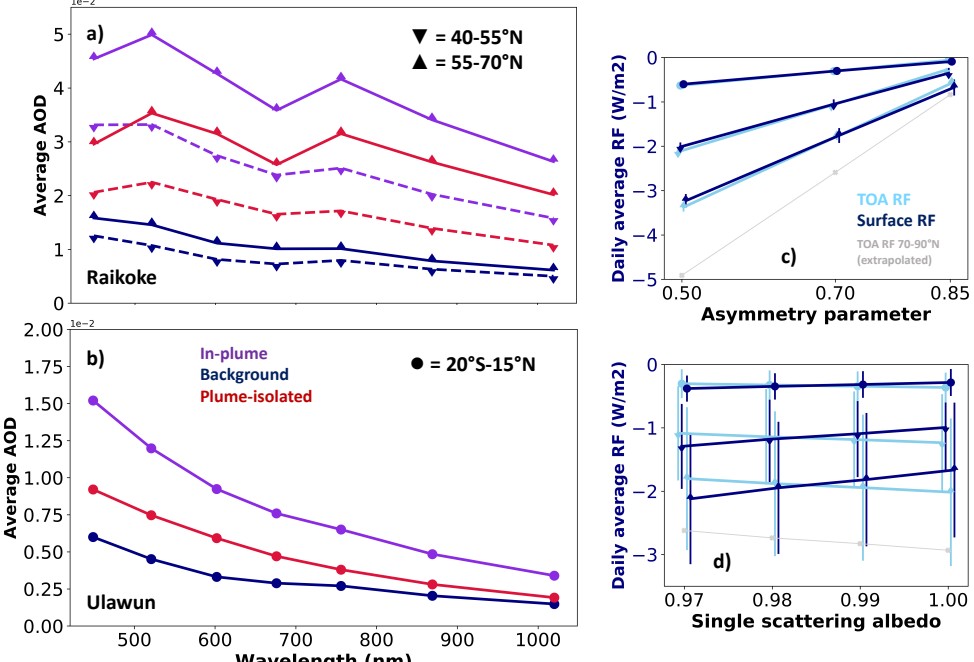

**Figure 8.** (a) Average SAGEIII/ISS stratospheric AOD vs. wavelength (from tropopause to TOA) attributed to Raikoke (average in the period from the eruption to end of September 2019, violet lines), corresponding background (September $1^{st}$ to $15^{th}$ 2018, blue lines) and Raikoke plume-isolated (Raikoke - background, red lines), in the latitude bands 40-55°N (downward triangle and dashed lines) and 55-70°N (upwards triangle and solid lines). (b) Average SAGEIII/ISS stratospheric AOD vs. wavelength attributed to Ulawun, second eruption, in August 2019 (average in the period from the eruption to end of November 2019), corresponding background (June $15^{th}$ to $30^{th}$ 2018, blue lines) and Ulawun plume-isolated (Ulawun - background, red lines), in the latitude bands 20°S-15°N (circles), same color code as panel a. (c) Equinox-equivalent clear-sky daily average radiative forcing, at TOA (sky blue symbols and lines) and surface (dark blue symbols and lines), as a function of the hypothesis on the asymmetry parameter (and averaged over all single scattering albedo hypotheses), for Ulawun (at 20°S-15°N, circles) and Raikoke (at 40-55°N, downward triangles, and at 55-70°N, upward triangles). An OMPS-based extrapolation of the radiative forcing at 70-90°N is also shown with grey crosses and lines. Error bars are a measure of the variability of the RF estimations with the different hypotheses on SSA. (d) Same as panel c but as a function of the assumptions on the single scattering albedo (and averaged over all asymmetry parameter hypotheses). Error bars are a measure of the variability of the RF estimations with the different hypotheses on the asymmetry parameter.

reference and have to be scaled down to take into account the impact of clouds in reducing the effective RF.

Based on the above mentioned regional clear-sky RF estimations in the shortwave (Table 2), the equinox-equivalent daily average shortwave global TOA radiative forcing of Raikoke and Ulawun plumes, based on their stratospheric aerosol layer perturbations, can be estimated. We calculated this as a latitude-weighted mean of the regional RF, extended over the whole globe, by considering a zero-impact outside the regions defined in this section. Because we know that the Raikoke plume had an influence on the tropics (which is here considered as a 'zero impact region'), the calculated global clear-sky RF values are



**Table 2.** Global clear-sky TOA RF estimations (in W/m$^2$). Experiment 1: using shortwave SSA between 0.97 and 1.0 and shortwave g between 0.50 and 0.85, Experiment 2: using shortwave SSA between 0.98 and 1.0 and shortwave g between 0.50 and 0.70.

|  | **Raikoke** | **Ulawun** |
|---|---|---|
| **Experiment 1** | -0.27±0.09 | -0.09±0.03 |
| **Experiment 2** | -0.38±0.06 | -0.13±0.02 |

likely underestimated. The clear-sky global averages are listed in Tab. 2, for Raikoke and Ulawun, and for two hypotheses: an average of all SSA and g hypotheses (Experiment 1) and excluding the extreme values of SSA (0.97) and g (0.85), which are linked to relatively large absorption and large average particles size (Experiment 2). Values as high as -0.38 W/m$^2$ are found

for Raikoke. The all-sky to clear-sky RF ratio for the Sarychev eruption has been reported as about 0.4 (Haywood et al., 2010); the Sarychev eruption occurred at a very similar period of the year and location with respect to Raikoke. Thus, we applied this empirical scaling factor, and obtained an all-sky RF of Raikoke in the range -0.11 to -0.16 W/m$^2$, which is very similar to the estimation for Sarychev (Haywood et al., 2010). Smaller values are found for Ulawun: a clear-sky RF of -0.09 to -0.13 W/m$^2$, extendable to values of -0.04 to -0.05 W/m$^2$ at all-sky conditions.

## 530 6 Conclusions

We show that during the past 3 years, the highest peak sAOD values resulted from the Raikoke eruption. This series includes the Canadian fires (2017), the Ambae eruption (2018) and the Australian fires in 2019/2020. During the eruption multiple plumes were injected on different altitudes at different times containing SO$_2$ and ash, making this eruption challenging for the modelling world. During the first few days after the eruption the Raikoke plume was entrained in the Aleutian cyclone. Within

3 weeks to one month after the Raikoke eruption, the plume has circled the Earth. Stratospheric AOD values as high as 0.045 (at 449 nm) and decreasing to about 0.04 (longer-wavelength visible, 676 nm) and 0.03 (near infrared, 1020 nm) are observed in higher NH latitudes, with an average 0.025 value at longer-wavelength (visible, 675 nm) in the NH. The background sAOD is still enhanced in the NH one year after the eruption. The OMPS aerosol extinction observations show a rising of aerosol-filled air masses from ∼15 km in July to 21 km in September from the Raikoke eruption. In the same period, a smaller impact from

the Ulawun eruptions, especially the one in August 2019, is also observed. The Ulawun plume circled the Earth in the tropics within one month and led to sAOD values of ∼ 0.01, in the visible, in the tropics. The Ulawun plume was mainly transported towards the South. A possible transport towards the North within the BDC is masked by already increased sAOD values from the Raikoke eruption in the NH. Even though SAGE III/ISS has a much sparser sampling rate than OMPS, the monthly sAOD evolution on broad latitudinal bands is reliably represented in terms of absolute value (in the tropics and NH) and timing for all

documented stratospheric aerosol events. Discrepancies (in terms of aerosol concentration and lifetime) between observations and the global model WACCM point to the complexity of those events. In particular it may indicate that the initial injection of ash (which is not implemented in the WACCM set up) plays a role in the evolution of such plumes, in particular for Raikoke. The global RF for Raikoke is estimated at values between -0.3 and -0.4 W/m$^2$, in clear-sky conditions and can be scaled to





values of -0.1 to -0.2 W/m$^2$ at all-sky conditions. Simulation results potentially indicate an impact of the Raikoke plume on the

SH. This would lead to an underestimation of the given global RF values. As is, our estimation is on par with or exceeding the RF of the well-studied Sarychev eruption in 2009, thus setting a new reference for climatic impacts of stratospheric aerosols perturbations for the post-Pinatubo-influenced period. The RF of the Ulawun eruptions is down to 4 times smaller than the one for Raikoke and is, in this respect, negligible.

*Data availability.* The aerosol extinction data sets from SAGE III-ISS v5.1 are available at https://eosweb.larc.nasa.gov and OMPS v1.5

at https://daac.gsfc.nasa.gov/. The model and simulation data may be requested from the corresponding author: the CLaMS model data (f.ploeger@fz-juelich.de), the UVSPEC input and output files for the radiative forcing calculations (pasquale.sellitto@lisa.u-pec.fr). Himawari-8 and IASI Level 1c data are provided by AERIS/ICARE data centre (https://en.aeris-data.fr/direct-access-icare/), the ERA5 data are available from Copernicus Climate Change Service (https://climate.copernicus.eu/climate-reanalysis). LOAC data are available at http://doi.org/10.5281/zenodo.3937477.





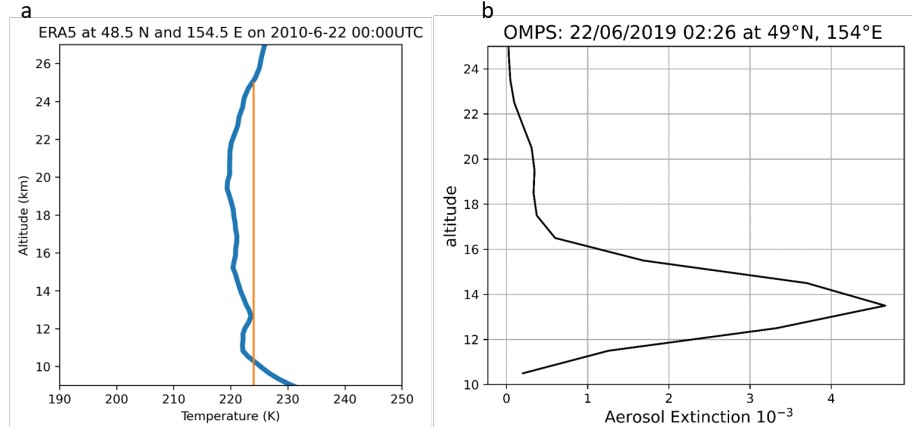

**Figure A1.** (a) ERA5 temperature profile (blue line) at the location of the minimum brightness temperature (orange line) of Himawari-8 observations of the Raikoke plume a few hours after the eruption. (b) OMPS aerosol extinction profile close to the Raikoke location shortly after the injection phase.

## Appendix A:  Supporting material for Section 4.1

Figure A1: Observations for the analysis of the injection of the Raikoke plume. This is used to determine the input of the WACCM initialization of the plumes' injection following the Raikoke eruption on 21-22/06/2019.

Figure A2: Same as for Fig. A1, but for the injection of the Ulawun plumes for both eruptions. Additionally, we present the corresponding Himawari Ash RGB, showing a clear signal of ash on August $3^{rd}$ for the second Ulawun eruption.

Movie: GIF of the Raikoke eruption from 18:00 UTC on June $21^{st}$ to 09:40 UTC on June $22^{nd}$ at 20 minutes interval (http://doi.org/10.5281/zenodo.3939167). Notice the series of explosions that occurred at many instances between 18:00 UTC and 5:40 UTC, in particular, the two last ones at 3:40 and 5:50 UTC. The images are produced using the RGB Dust recipe like Fig. 1.

Figure A3: IASI observations show the entrainment of $SO_2$ enhanced air masses in the cyclonic circulation of the Aleutian low.

*Author contributions.*  C.K., P.S., B.L. and G.B. designed the research, analyzed and interpreted data. C.K. carried out the OMPS and SAGE III/ISS analysis. B.L. carried out the Himawari data analysis. M.E. and P.S. produced the IASI $SO_2$ detection observations. G.B. and M.T. carried out the WACCM simulations. F.P. carried out the CLAMS simulations. P.S. carried out the radiative forcing calculations and related analyses. G.T. provided expertise on OMPS data. JB.R. and G.B. provided LOAC observations, data treatment and analysis. F.J., G.T. and A.B. were involved in the discussions. C.K. wrote the paper with contributions from P.S., G.B. and B.L.. All authors approved the final version.





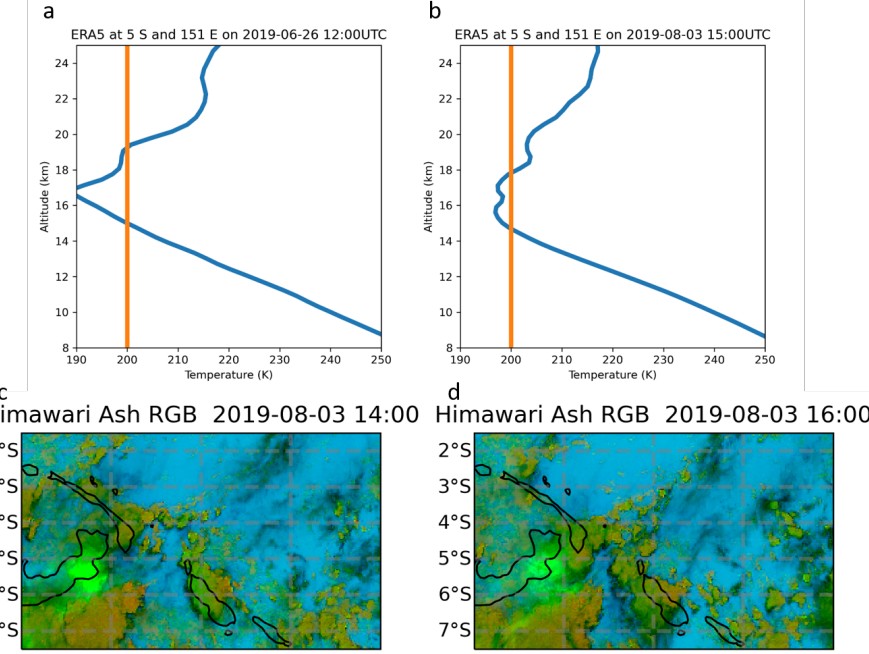

**Figure A2.** (a) and (b) as in Fig. A1 (a), but for both Ulawun eruptions accordingly. (c) and (d) Similar to Figure 1, Himawari ash RGB for the second Ulawun eruption. Light green represents ash, while darker green shades show clouds.

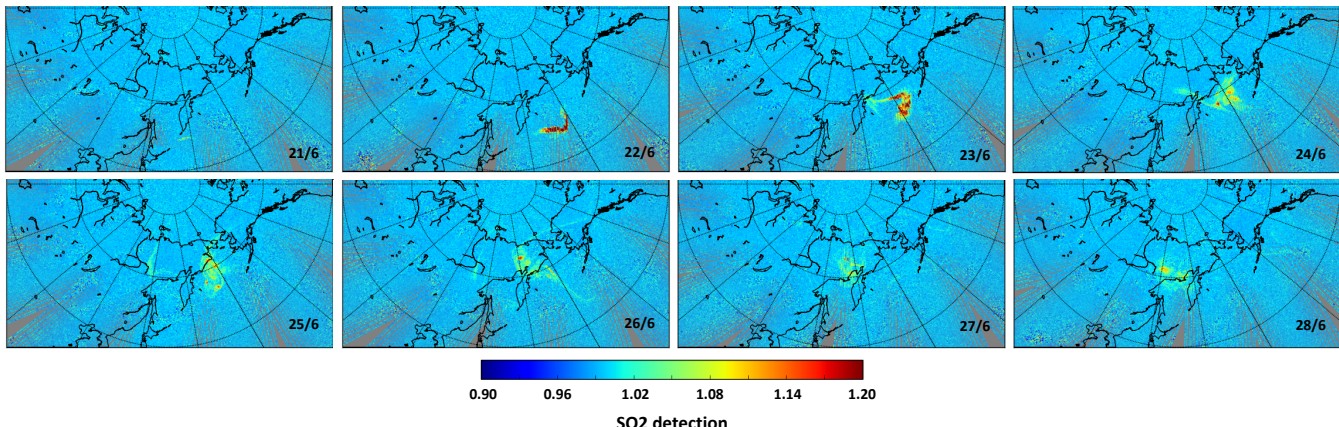

**Figure A3.** IASI SO$_2$ observations from June $21^{st}$ to $28^{th}$.

*Acknowledgements.* The authors are thankful for the support of ANR (Agence Nationale de La Recherche) under grant ANR-17-CE01-0015 (TTL-Xing) and ANR-10-LABX-100-01 (French Labex VOLTAIRE managed by University of Orleans). C.K. was funded by Deutsche Forschungsgemeinschaft (DFG, German Research Foundation) - 409585735. F.P. was funded by the Helmholtz Association under grant VH-NG-1128 (Helmholtz Young Investigators Group A-SPECi). The authors acknowledge the providers of LibRadtran and the National





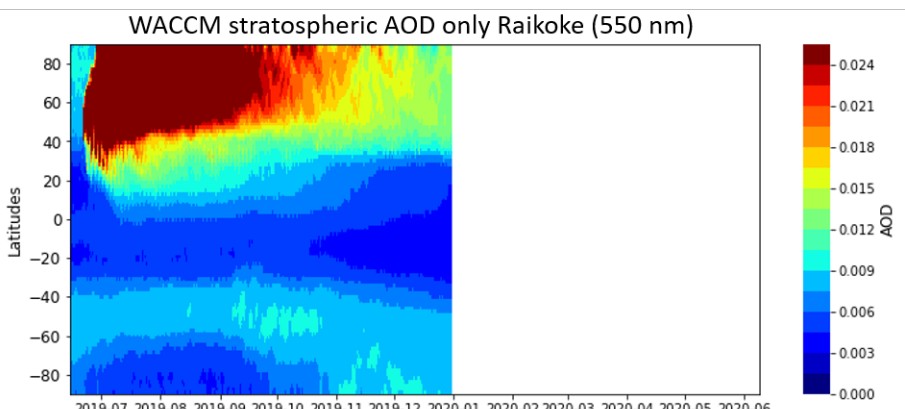

**Figure A4.** The integrated stratospheric column for the WACCM simulation for the Raikoke eruption, respective to Fig. 4b and c.

580     Aeronautics and Space Administration (NASA), the SAGE III/ISS and OMPS teams. The IASI-related activities were supported by CNES (TOSCA IASI project). The authors are grateful to Tong Zhu from SSAI and Talat Khattatov for technical support.





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
