# Peer review of "Stratospheric aerosol layer perturbation caused by the 2019 Raikoke and Ulawun eruptions and their radiative forcing"

_Atmospheric Chemistry and Physics, 2020_

## Referee Comment (RC1) · Anonymous Referee #1 · 3 Sep 2020

Comment on "Stratospheric aerosol layer perturbation caused by the 2019 Raikoke and Ulawun eruptions and climate impact" by Kloss et al.

General comments:

Kloss et al. study the change of the stratospheric aerosol layer after the eruption of Raikoke and Ulawun in 2019, and also try to address the climate effects of the eruptions. The main data sources are satellites and several in situ AOD measurements. And they used the CLaMS model to study the transport of the "hypothetical" volcanic plume, and they used the WACMM model do estimate the radiative forcing possibly produced by the two volcanic eruptions. Generally, the manuscript is nicely structured and the study may provide useful information for researchers interested in volcanism and stratospheric aerosols. I can see the authors have done a lot of work, collecting comprehensive data and running multiple models, for this study. However, the manuscript seems to be composed in a rush and many technical corrections are needed.

Also, I have a few questions that need to be addressed before the manuscript could be published.

My general questions are as follows:

1. The title.

I see the WACCM model results provide TOA radiative forcing. However, TOA radiative forcing does not equal to "climate impact". To be accurate and avoid misleading information, I would prefer to use "radiative forcing" instead of "climate impact" in the title.

2. The IASI based $D_{so2}$.

Did you verify your definition of SO2 concertation in a previous study? If yes, please cite it. If no, please verify the definition and comment on the performance before use.

3. The LOAC data.

Why are the uncertainties explained here different from and worse than the uncertainties in Section 2.3 in Renard et al. (2016)? I assume you used a newer model of LOA

I assume you calculate stratospheric AOD (sAOD) from the LOAC data above the tropopause to 23 km. But you did not make it clear in the manuscript.

4. I would suggest you move section 3, the introduction of the two eruptions, to a more appropriate location, before the CLaMS and WACMM model setting, because it is better to know the date of the eruption, plume height, SO2 volume, etc. before the model set. After moving section 3, please also check the texts and remove the overlapped information of the eruptions in section 3 and in the model setting section.

5. The CLaMS simulation of the dispersion of volcanic plume.

The authors know it very well that the initial plume box for the CLaMS simulation is not accurate, so the simulation results are only suitable for a rough estimation. But this rough assumption would also make the simulation not very necessary.

In about half a month after eruptions, the SO2 concentration and SO2 plume height would be a nice proxy for volcanic plume dispersion, as you showed in Fig. A3. Or as

in https://iasi.aeris-data.fr/so2/. For a longer time after the eruptions, the error of trajectories accumulates and the results are even more unreliable. Are the CLaMS results in Fig.5 supported by the OMPS in Fig. 3 or WACCM simulations in Fig. 4? If not, it would be better to only keep reliable results.

5. Figure 2

There are very small Dso2 values in the figures, such as in Fig.2a, bottom left corner in Fig.2c, and bottom right corner in Fig.2d. They are probably not SO2 from Raikoke. They may be removed if you only show data with large signal/noise ratios.

**Specific comments and corrections:**

Please make the font of the manuscript uniform.

Please read the manuscript multiple times to correct typos. I list some of them but not all of them here.

**Page 1**

    L1 a moderate stratospheric eruption;

    L4 short-wave length, high northern …

    L6 evolution of what?

    L14 RF. Please spell it out when you use the abbreviation for the first time.

**Page 2**

    L21–22 please unify the format of brackets (all half or all full).

    L28 Brewer-Dobson circulation

    L31 jets

    L42 "0.7-2.2" (and many other places in the manuscript): please find out the differences between hyphens and dashes and use them right.

    L46 Fromm et al. … This sentence is very confusing, please try to rephrase.

**Page 3**

    L58 setup

    L69 "1.5 (Rault and Loughman, 2013)."

    L71 )) Please add reference to MERRA2 data here.

    L72 Are the OMPS AOD data contaminated by ordinary clouds?

    L74 L81 dataset

    L82 +-   --> ±

**Page 4**

    L109 "nm" is not a unit for wavenumber. I guess you may want to say "cm$^{-1}$".

**Page 4**

    L85 "marked added-value" Can't understand.

    L95 "The Dust RGB product performs better for volcanic plumes than the Ash RGB product at large viewing angles." Please add a reference here, or explain it if a reference is not available.

**Page 6**

    L150 "- the" L153 "- The" L154 "- The": I do not understand the usage of hyphens.

L152 "Mid-latitude"

**Page 8**

L238 moving

L239 usually

**Page 25**

Please put Fig. A1 together with other figures in the appendix.

Please put brackets around "$10^{-3}$" in Fig.A1b to make the X label format uniform.

And add (km) as Y label for Fig.A1b.

The font size of figure titles is not the same.

**Page 26**

Please add a latitude range for Fig. A2c–d.

Please put the acknowledgement together with other texts.

**Page 28**

Please double check the format of your references ONE BY ONE to make sure they are in the ACP reference format.

**Page 30**

Please avoid citing a paper that you are not sure whether it is finished or not.

---

## Referee Comment (RC2) · Anonymous Referee #2 · 3 Sep 2020

**Review of: Stratospheric aerosol layer perturbation caused by the 2019 Raikoke and Ulawun eruptions and climate impact by Corinna Kloss et al**

There are some very nice aspects of this paper – particularly the OMPS analysis and the WACCM model simulations.

Generally, I was less impressed with the CLAMS modelling which has to be coupled to an off-line RT code (UVSPEC) and SAGEIII aerosol profiles in order to provide a rough estimate of the radiative effect. The radiative effect will only ever be rough because i) there are no clouds considered in the RT calculations, ii) how variations in the surface albedo were taken into account is not documented, iii) it makes a simple equinox assumption in order to simplify impacts from daylength, solar zenith angle variations etc. It would have been a better approach to examine the effective radiative forcing in the WACCM model and adjust some of the optical parameters within that model as this would have allowed for precise co-location between e.g. aerosol, cloud, surface reflectance, atmospheric profiles etc.

It does seem as though it has been rather rushed – particularly the latter sections lack the methodological rigor to provide a credible estimate of the climate impact via the radiative forcing. There are some areas where the text and Figure captions needs clarifying/improving. The choice of Figure presentation is not always optimal.

On balance, I feel that the paper does provide enough new results to be of interest to the scientific community, but I would recommend that the authors consider both the more major comments and typos/corrections before the paper is acceptable for publication.

**More major Comments:**

**Abstract: "**Discrepancies between observations and models indicate that ash has played a role on evolution and sAOD values."

This is rather overstating what you conclude in your main text in your conclusions: "Discrepancies (in terms of aerosol concentration and lifetime) between observations and the global model WACCM point to the complexity of those events. In particular it may indicate that the initial injection of ash (which is not implemented in the WACCM set up) plays a role in the evolution of such plumes, in particular for Raikoke."

It is important therefore to change "ash has played" to "ash may have played"

**L223: "**There are no CALIOP intersections of the core plume during the early stage". I would contest this. There is an overpass on the 22nd from what I can see. Have a look at the data here:

https://www-calipso.larc.nasa.gov/products/lidar/browse_images/show_v4_detail.php?s=production&v=V4-10&browse_date=2019-06-22&orbit_time=15-59-28&page=1&granule_name=CAL_LID_L1-Standard-V4-10.2019-06-22T15-59-28ZN.hdf

While not extensive (and it shouldn't be owing to the narrowness of the plume at that time), at ~ 50N, there is evidence of stratospheric aerosol at 16-17km. From the website, the potential temperature looks to be around 425K at 100hPa. A rough conversion to temperature gives me 220K. This is pretty close to your 225K. I would therefore suggest a slightly more rigorous assessment would be worthwhile using this CALIPSO data given that these values tend to support your assumptions. This will give the reader more confidence that your assumptions are robust.

**L233 and Caption Figure 1:** There are some inconsistencies between the text and the figure caption: Text: "This plume is initially composed of ash (reddish colors, in Fig. 1), with also some evidence of SO2 (yellowish colors, in Fig. 1). The remaining greenish and pinkish colors indicate the presence of water clouds around the volcanic plume." Caption ". Red: ash; Pink to violet: dust; Yellow: mixture of ash and SO2, Green: thick and thin mid-level clouds or cirrus clouds". What is the difference between ash and dust in your caption? While I recognise that these are semi-quantitative estimates, the text should be better reconciled. The imagery is always semi-quantitative in the absence of in-situ observations of the ash owing to e.g. different refractive indices giving different 'colors' even for the same size distribution (e.g. Figure 6 of Millington et al., 2012 which uses the SEVIRI dust product; reference provided below). Some caveats surrounding this identification should be given and Millington et al. (2012) or similar should be referred to.

Millington, S. C., Saunders, R. W., Francis, P. N., & Webster, H. N. (2012). Simulated volcanic ash imagery: A method to compare NAME ash concentration forecasts with SEVIRI imagery for the Eyjafjallajökull eruption in 2010. *Journal of Geophysical Research: Atmospheres*, *117*(D20).

**The discussion starting around line 300:** "For both Ulawun eruptions (June 26th and August 3rd), OMPS data show some AOD perturbations after the first eruption and more significantly elevated values after the second eruption. Like for the Raikoke eruption, WACCM shows immediate and stronger signals during the weeks following the eruptions, but decreasing faster. While for OMPS observations a significant impact (sAOD around 0.01) of the second Ulawun eruption is still apparent in the tropical stratosphere by the end of the year 2019, in the model comparable values are found in October and by the end of 2019 the sAOD has values down to 10 times smaller than for OMPS. The model shows a faster decrease…."

These statements would be aided by the addition of simple line plots of the global and hemispheric sAODs. Figure 7a does show OMPS sAODs integrated over some latitude bands in such a manner, over a longer time period. However, I think that it would be worthwhile indicating the global, 30-90N, 0-30N, 0-30S, etc for both the model and the observations as a comparison.

**CLAMS model:** The initialisation of the model is pretty coarse (a box) which doesn't have the details of the spatial distribution in the vertical or horizontal within the plume. More care is therefore needed in interpreting the results from the CLAMS model. For example, "Therefore, the sAOD enhancement above Europe observed by OMPS in Fig. 3b does not originate from Raikoke, but rather from forest fires in Alberta, Canada."

"Europe" is a large area: The OMPS data suggests that there is an enhancement of the AOD over northern Europe, Scandinavia, the Baltic countries, and western Russia (Fig 3). The CLAMS simulations suggest that the Raikoke plume impacts "southern" Europe. Areas such as the UK at the interface between northern and southern Europe experience both …… Some of this greater detail is worth stating more explicitly, plus the caveat that the CLAMS initialisation may not be that accurate.

**Section 4.4. Vertical distribution**. While most of the graphical displays are reasonably logically chosen throughout, here I think that the choice of representation of the vertical distribution could be improved. Figure 6a-d are "around Raikoke and around Ulawan", while Figure 6e shows the OMPS data in a series of time stamps as a function of latitude and altitude. I would have preferred to see the model distributions plotted up in a similar way to the Fig 6e. One could then see if the modelled aerosol plumes interact or overlap (probably more likely) from using either the WACCM model or the CLAMS model. The approximate location of the stratosphere could be marked on Figure 6e and any of the new figures too.

**Section 5: Radiative effects:** "calculated the shortwave RF of the Raikoke and Ulawun plumes using the UVSPEC radiative transfer model (see Sect. 2.7 for the setup of the model and calculations). As input parameters for the model, the SAGE III/ISS volcano-attributed aerosol extinction profiles discussed above are used."

Why isn't the radiative forcing (or the effective radiative forcing) given for the WACCM model? It can be used for these calculations can't it? The use of the SAGE extinction profiles and sensitivity perturbations of the single scattering albedo allow some assessment of the impacts on the clear-sky radiative forcing and the sensitivity to the assumptions. WACCM should be able to give both cloud-free and cloudy sky effective radiative forcings but these are absent from the paper.

How is the surface reflectance taken into account? I could not find details. Won't the co-location of the highest AODs over the highest surface reflectances need to be accounted for (weakening the TOA radiative forcing)?

**Typos/clarifications:**

The level of English is generally acceptable, but there are a number of corrections noted below that will make the paper easier to read and digest. I would suggest that a native English speaker re-read the amended manuscript before re-submission as I won't have caught all of them.

l1: stratospheric moderate -> moderate explosive

l15: Suggest Severe -> Explosive

l17: of sulfur dioxide (SO2) volcanic emissions -> volcanic emissions of sulphur dioxide (SO2)

l23: dominates -> strongly influences. You cannot say that it dominates as if it were an effusive eruption emitted at the surface it would have littleclimate effect (except perhaps through aerosol-cloud-interactions)

l28: Butchart, 2014 -> Butchart, 2014; Jones et al., 2017. I think that the study by Jones et al (2017) is worth including here. Their Figure 1, is perhaps one of the most relevant in terms of the injection latitude and altitude.

Jones, A.C., J.M. Haywood, N. Dunstone, M.K. Hawcroft, K. Hodges, A. Jones, and K. Emanuel, Impacts of hemispheric solar geoengineering on tropical cyclone frequency, Nature Communications, 8, 1382, doi:10.1038/s41467-017-01606-0, 2017.

l29: Point (3) does not have a suitable reference associated with it. I would suggest adding the Jones et al (2017) reference again here (see above): relative to the tropopause -> relative to the tropopause (e.g. Jones et al., 2017)

l 33: 20Tg SO2 is quite a large estimate for the amount of SO2 injected. I would suggest "Up to around 20Tg SO2"

l34: have been -> were

l37: climate occurred -> climate has occurred

l43: its good practice to be sequential in terms of the dates: Günther et al., 2018; Kristiansen et al., 2010; Krotkov et al., 2010 -> Kristiansen et al., 2010; Krotkov et al., 2010; Günther et al., 2018

l51: the complexity that -> the complexity and the uncertainty that

l53: time the -> time, the

l54: Canada, Alberta (June) and Siberia (July) -> Alberta, Canada (June) and Siberia, Russia (July)

l64: flies -> has flown

l85 on multiple -> at multiple: Agreed: is it worth saying explicitly that the wavelength dependence provides information on the aerosol size distribution via the Angstrom exponent?

l93: to discriminate -> discrimination between

104: "volcanic effluents" is a strange phrase: I'd replace with "emitted in volcanic plumes".

L116: micronic -> micron

L147: With the UVSPEC the -> With UVSPEC, the

L150-l155: remove the "-"s for grammar purposes.

L191: for a pure -> from a pure

L212: possibly refer again to Jones et al. (2017)

L217: as in -> to

L239: mowing -> moving

L244 & 247: The use of possibly is questionable. It definitely is converted to sulfate aerosol owing mainly to gas phase oxidation. Remove possibly in both sentences.

L262: A reference to the smoke from the Alberta fires would be appropriate. There may be better ones appearing at present, but here is one I found: Jenner, L.: Alberta Canada Experiencing an Extreme Fire Season, NASA, May 30, https://www.nasa.gov/image-feature/goddard/2019/ 545 alberta-canada-experiencing-an-extreme-fire-season, 2019.

Fig 3: Caption – the wavelength for the AOD should be stated.

L274. even one year -> even nearly one year

L279: The eastward transport dominates, which depends on the vertical distribution of the aerosol and the phase of the QBO (quasi-biennial oscillation). The sentence could do with a reference e.g. Lee and Smith, 2003:

Lee, H. and Smith, A.K., 2003. Simulation of the combined effects of solar cycle, quasi-biennial oscillation, and volcanic forcing on stratospheric ozone changes in recent decades. *Journal of Geophysical Research: Atmospheres*, *108*(D2).

L308: crossed-impact -> cross-impact

L325: interfered with the Raikoke evolution -> interfered with the evolution of the Raikoke plume

L334: mentioned limitations -> associated limitations

L340: which is a schematic estimate, but for sure causes discrepancies compared to observations and reality -> "which is a necessary simplification of reality where pulses in injection altitude and magnitude are inevitable".

L347: potential cloud signatures are included -> cloud signatures are potentially included

L357 locationsof -> locations of

Section Heading: "Recent" is a subjective term: Kasatochi/Sarychev could be considered to be recent. I would simply add the range of recent to the title "In the context of other recent events (2017-2020)"

Fig 7. I like Figure 7. It is very informative. As a minor point, it would have been more logical for the LOAC points to have been plotted in purple so that the latitude of the observations correspond to the latitude band in Fig 7a-b.

L451. The slight increase in the observed AOD in April 2019 -> The slight increase in the observed AOD in the southernmost latitude band in April 2019

Fig 8. The 1e-2 scaling on the ordinate axis is tiny! This really needs to be more clear.

---

## Author Comment (AC1) · 4 Nov 2020

*Referee #1*

*We would like to thank Reviewer 1 for his/her comments, which helped us improve the quality of the manuscript, as well as his/her fast response during the discussion phase. We discussed each of the points raised by Reviewer 1 among the coauthors and made the changes in the text accordingly. Below each comment you find our answers and the respective changes made.*

**1. The title.**

I see the WACCM model results provide TOA radiative forcing. However, TOA radiative forcing does not equal to "climate impact". To be accurate and avoid misleading information, I would prefer to use "radiative forcing" instead of "climate impact" in the title.

The title was changed to: *'Stratospheric aerosol layer perturbation caused by the 2019 Raikoke and Ulawun eruptions and their radiative forcing'*

**2. The IASI based Dso2.**

Did you verify your definition of SO2 concertation in a previous study? If yes, please cite it. If no, please verify the definition and comment on the performance before use.

We have extended Sect. 2.4, including a new figure, to better explain the definition of our $D_{SO2}$ parameter, which is a new parameter introduced in the present manuscript. Please note that $D_{SO2}$ is not a measurement of the concentration of $SO_2$ but a very simple "band-difference" to identify IASI pixels where a strong presence of the $SO_2$ absorption signature can be found. This is in no way a quantitative parameter and is only useful in cases, like the one described here, where a strong $SO_2$ emission is observed, with the only aim to observe the $SO_2$ plume dynamics. This is discussed now in the text.

*'…R(n) represents the radiance observed from IASI at the wavenumber n. The two values n1= 1129.25 nm and n2= 1130.25 nm represent two spectrally-close wavenumbers, the first at the center of a $SO_2$ absorption line and the second outside. Figure 1 shows a case of simulated IASI spectra with and without $SO_2$ (all other parameters in the simulations of the IASI spectra are the same, e.g. surface temperature, temperature and humidity profiles, gaseous absorbers and aerosol profiles). The two selected wavenumbers n1 and n2 are highlighted to show their extreme position (n1 at the approximate center and n2 outside the absorption feature) in one isolated $SO_2$ absorption line, which is not affected by the absorption of water vapor or other extra-$SO_2$ species. From the definition of Eq. 1 and Fig. 1 it is possible to see that values of $D_{SO2}$ larger than 1.0 are linked to spectra where $SO_2$ is detected. It is important to stress that $D_{SO2}$ is purely a qualitative detection parameter is not to be taken as a quantitative retrieval of the $SO_2$ concentration, even if linked to this latter. This parameter is only useful in case of strong $SO_2$ anomalies, like the one generated by the Raikoke eruption, and for the analysis of relatively large-scale dispersions of $SO_2$-rich plumes.'*

[Figure]

*Figure 1: Simulated IASI spectra with (black) and without (red) SO₂ and a zoom of the SO₂ absorption line used to define the $D_{SO2}$ parameter of Eq. 1.*

**3. The LOAC data.**
Why are the uncertainties explained here different from and worse than the uncertainties in Section 2.3 in Renard et al. (2016)? I assume you used a newer model of LOA
I assume you calculate stratospheric AOD (sAOD) from the LOAC data above the tropopause to 23 km. But you did not make it clear in the manuscript.

We thank the reviewer for their remark which has revealed a mistake in an uncertainty number we provide. The uncertainty is indeed ±20% for concentrations higher than 1 particle.cm$^{-3}$ and not 10 particles.cm$^{-3}$, which is in agreement with the Renard et al. AMT 2016 paper. We have refined the uncertainty values provided in the AMT paper by adding a specific information for submicronic particles; that is why we had written: the uncertainty increases to about ±30% for submicronic particle concentrations higher than 1 particle cm$^{-3}$. We have corrected the mistake in the new version of the text: *'It provides particles number concentrations for 19 sizes in the 0.2 – 50 µm size range, with an uncertainty of ±20% for concentrations higher than 1 particle.cm$^{-3}$'*

Second point: This is true. We changed the Figure caption accordingly *'Derived partial sAODs for balloon borne LOAC aerosol concentration observations from Ury in France, for particle sizes from 0.2-0.7 µm from the tropopause up to 23 km altitude'*

**4.** I would suggest you move section 3, the introduction of the two eruptions, to a more appropriate location, before the CLaMS and WACMM model setting, because it is better to know the date of the eruption, plume height, SO2 volume, etc. before the model set. After moving section 3, please also check the texts and remove the overlapped information of the eruptions in section 3 and in the model setting section.

Prior to submitting the manuscript, the authors changed and discussed the position of this section several times. We will move section 3 back before the methods section/ after the introduction.

**5. The CLaMS simulation of the dispersion of volcanic plume.**
The authors know it very well that the initial plume box for the CLaMS simulation is not accurate, so the simulation results are only suitable for a rough estimation. But this rough assumption would also make the simulation not very necessary.
In about half a month after eruptions, the $SO_2$ concentration and $SO_2$ plume height would be a nice proxy for volcanic plume dispersion, as you showed in Fig. A3. Or as in https://iasi.aeris-data.fr/so2/. For a longer time after the eruptions, the error of trajectories accumulates and the results are even more unreliable. Are the CLaMS results in Fig.5 supported by the OMPS in Fig. 3 or WACCM simulations in Fig. 4? If not, it would be better to only keep reliable results.

The dispersion of the plume that is simulated with a large number of trajectories follows essentially the evolving analyzed pattern of the atmospheric circulation that is much more reliable than individual trajectories. Numerous previous studies of transport in the lower stratosphere showed that plumes can be predicted one month ahead.
It should be also kept in mind that ClaMS is not just a trajectory model but also includes small-scale mixing processes (parameterized depending on the deformation rate in the large-scale flow). Hence, individual trajectories are only calculated over 24 hours (the mixing, or regridding, time step). The reviewer is, of course, right that quantitative comparison between the observations and the simulation without microphysics included is difficult. Our intention when including ClaMS simulation in the paper was to more qualitatively illustrate the pure effect of passive transport on the plume. And the comparison to OMPS indeed shows that passive transport explains the large-scale dispersal of the plume quiet well. However, we weakened the respective statements in the revised manuscript, in order not to overemphasize the comparison between ClaMS and OMPS too much. Relevant sentences are:
*'The plume air mass transport is* qualitatively  *consistent with OMPS observations, as by the end of July (Fig. 4) enhanced AOD values are apparent throughout all longitudes, mostly north of the Raikoke position. For the CLaMS simulation a clear signal of the tracer is visible around the area of the AMA from end-July until mid-September, which is also consistent with OMPS data (Fig. 4c-e). By mid-August a small percentage of the initialized Raikoke tracer has reached the tropics in the CLaMS simulations…'* And the last sentence of this section *'Even though CLaMS simulations neither take any chemical/microphysical processes into account nor possible lifting due to aerosol-radiation-dynamics (suggested to play a crucial role for the Raikoke eruption in Muser et al. (2020)), comparisons show that the horizontal passive tracer distribution from the ClaMS simulation illustrates the effect of passive transport for plume dispersal.'*

**5. Figure 2**
There are very small Dso2 values in the figures, such as in Fig.2a, bottom left corner in Fig.2c, and bottom right corner in Fig.2d. They are probably not SO2 from Raikoke. They may be removed if you only show data with large signal/noise ratios.

*As discussed now in the text in Sect. 2.4 and in the reply to major comment 2, $D_{SO2}$ is a purely band-difference detection algorithm, very useful in terms of large-scale analyses of the dispersion of $SO_2$-rich plumes but not expected to be very accurate at smaller scales. Probably, the small values pointed out by the Referee are false detection due to other spurious spectral signatures (surface emissivity, high clouds, other infrared-radiation-absorbing species or, of course, $SO_2$ from other sources), which are impossible to filter-out based on retrieval performances. In any case, it is not critical to discuss the first phases of the large-scale dispersion, based on $D_{SO2}$, which has a clear signature in the Raikoke plume due to the very high $SO_2$ concentrations in the initial phase.*

**Specific comments and corrections:**
Please make the font of the manuscript uniform.
Please read the manuscript multiple times to correct typos. I list some of them but not all of them here.

**Page 1**
L1 a moderate stratospheric eruption; *we avoided the term 'moderate now'*
L4 short-wave length, high northern … *ok*
L6 evolution of what? *'has influenced the extent and evolution of the sAOD'*
L14 RF. Please spell it out when you use the abbreviation for the first time. *ok*

**Page 2**
L21–22 please unify the format of brackets (all half or all full). *Thank you*
L28 Brewer-Dobson circulation
L31 jets
L42 "0.7-2.2" (and many other places in the manuscript): please find out the differences between hyphens and dashes and use them right.
*Thank you. This has been done accordingly (not marked in yellow).*
L46 Fromm et al. …This sentence is very confusing, please try to rephrase.
*"Fromm et al. (2014) raise awareness  data quality , but also conflicting injection sequence information  can lead to different conclusions about the same volcanic eruption."*

**Page 3**
L58 setup *ok, changed for all cases*
L69 "1.5 (Rault and Loughman, 2013)." *ok*
L71 )) Please add reference to MERRA2 data here. *ok*
L72 Are the OMPS AOD data contaminated by ordinary clouds? *Clouds are not filtered for the data set that we use. However, we focus on the stratosphere and therefore clouds do not play a significant role for this study. A sentence has been added for clarification: "To avoid removing enhanced aerosol layers that were mistakenly identified as clouds, we use the unfiltered OMPS dataset. The influence of stratospheric clouds*

*for the interpretation of this transport study about the Australian fire plume is expected to be negligible and not further analyzed."*

L74 L81 dataset

L82 +- --> ± *ok*

**Page 4**

L109 "nm" is not a unit for wavenumber. I guess you may want to say "cm–1". *Thank you*

**Page 4**

L85 "marked added-value" Can't understand.

*"However, the better vertical resolution and observations on multiple wavelengths compared to OMPS,* bring an *added-value when spatio-temporally averaged data are used for the radiative forcing calculations."*

L95 "The Dust RGB product performs better for volcanic plumes than the Ash RGB product at large viewing angles." Please add a reference here, or explain it if a reference is not available. *Reference added: Eumetrain 2020*

**Page 6**

L150 "- the" L153 "- The" L154 "- The": I do not understand the usage of hyphens. *This was supposed to represent a sequence. We changed this to numbering.*

L152 "Mid-latitude" *ok*

**Page 8**

L238 moving *ok*

L239 usually *no -> exceptionally*

**Page 25**

Please put Fig. A1 together with other figures in the appendix. *It is the ACP Latex template, which places the Figures. So we hope/believe that this will be handled during the formatting step by ACP.*

Please put brackets around "10–3" in Fig.A1b to make the X label format uniform. *ok*

And add (km) as Y label for Fig.A1b. *ok*

The font size of figure titles is not the same. *This has been changed for all Figures in the Appendix accordingly.*

**Page 26**

Please add a latitude range for Fig. A2c–d. *ok*

Please put the acknowledgement together with other texts. *Latex ACP template*

**Page 28**

Please double check the format of your references ONE BY ONE to make sure they are in the ACP reference format. *This is a typesetting issue. More information are given in the original Latex file. Copernicus chooses which information to use for their style.*

**Page 30**

Please avoid citing a paper that you are not sure whether it is finished or not.

*By now the Khaykin paper is published, the reference is changed accordingly.*

---

## Author Comment (AC2)

*We would like to thank Reviewer 2 for the time he/she spent on the detailed and mostly positive comments and suggestions (including an independent, additional literature search), to improve our manuscript! In the following, we address each comment individually, including the changes we made to the manuscript accordingly.*

**1) Abstract:** "Discrepancies between observations and models indicate that ash has played a role on evolution and sAOD values."
This is rather overstating what you conclude in your main text in your conclusions: "Discrepancies (in terms of aerosol concentration and lifetime) between observations and the global model WACCM point to the complexity of those events. In particular it may indicate that the initial injection of ash (which is not implemented in the WACCM set up) plays a role in the evolution of such plumes, in particular for Raikoke."
It is important therefore to change "ash has played" to "ash may have played"
The sentence was changed to: *'Discrepancies between observations and models indicate that ash* may have influenced the extent and evolution of the sAOD'

**2) L223:** "There are no CALIOP intersections of the core plume during the early stage".
I would contest this. There is an overpass on the 22nd from what I can see. Have a look at the data here: https://www-calipso.larc.nasa.gov/products/lidar/browse_images/show_v4_detail.php?s=production&v=V4-10&browse_date=2019-06-22&orbit_time=15-59-28&page=1&granule_name=CAL_LID_L1-Standard-V4-10.2019-06-22T15-59-28ZN.hdf
While not extensive (and it shouldn't be owing to the narrowness of the plume at that time), at ~ 50N, there is evidence of stratospheric aerosol at 16-17km. From the website, the potential temperature looks to be around 425K at 100hPa. A rough conversion to temperature gives me 220K. This is pretty close to your 225K. I would therefore suggest a slightly more rigorous assessment would be worthwhile using this CALIPSO data given that these values tend to support your assumptions. This will give the reader more confidence that your assumptions are robust.

*Yes, we are aware of this intersection. With the following Figure it becomes clear that CALIOP did not intersect the bulk of the cloud seen from HIMAWARI, but rather a very thin tail. Therefore, we believe the CALIOP observations not to be of sufficient value for this study.*

[Figure]

**3) L233 and Caption Figure 1:** There are some inconsistencies between the text and the figure caption: Text: "This plume is initially composed of ash (reddish colors, in Fig. 1), with also some evidence of SO2 (yellowish colors, in Fig. 1). The remaining greenish and pinkish colors indicate the presence of water clouds around the volcanic plume." Caption ". Red: ash; Pink to violet: dust; Yellow: mixture of ash and SO2, Green: thick and thin mid-level clouds or cirrus clouds". What is the difference between ash and dust in your caption? While I recognise that these are semi-quantitative estimates, the text should be better reconciled. The imagery is always semi-quantitative in the absence of in-situ observations of the ash owing to e.g. different refractive indices giving different 'colors' even for the same size distribution (e.g. Figure 6 of Millington et al., 2012 which uses the SEVIRI dust product; reference provided below). Some caveats surrounding this identification should be given and Millington et al. (2012) or similar should be referred to.

Millington, S. C., Saunders, R. W., Francis, P. N., & Webster, H. N. (2012). Simulated volcanic ash imagery: A method to compare NAME ash concentration forecasts with SEVIRI imagery for the Eyjafjallajökull eruption in 2010. Journal of Geophysical Research: Atmospheres, 117(D20).

We have changed the text accordingly (Figure 1 is now Figure 2):
*"The Dust RGB product is used, instead of the Ash RGB product, because it is more sensitive for large satellite viewing angles, which is the case for the region of interest for Raikoke. This product is based on the stronger absorption of ashes at 12 µm than at 10.4 µm while it is the opposite for ice and liquid water and on the absorption by $SO_2$ at 8.7 µm. It depends a lot on the size distribution of aerosols and ice crystals and provides only qualitative information (Millington et al., 2012). This plume is initially composed of ash (reddish colors, in Fig. 2), with also some evidence of $SO_2$ (yellow and bright green colors, in Fig. 2). The remaining brownish and blueish colors indicate the presence of water and ice dominated clouds associated to the volcanic plume."*

*Furthermore, we changed the caption for Figure 2: "Himawari Dust RGB images from 21/06/2019 to 28/06/2019, over the region of Raikoke. Red: ash; Bright green: $SO_2$; Yellow: mixture of $SO_2$ and thin ash; Greenish: thick and thin mid-level clouds or cirrus clouds; Brown: thick and high ice clouds; Blue: humid low level air; Pink to violet: dry low level air. The contour lines are plotted …"*
*As well as for Figure A2: "Himawari ash RGB for the second Ulawun eruption. Bright green represent $SO_2$, while darker green shades show clouds"*

**4) The discussion starting around line 300:** "For both Ulawun eruptions (June 26th and August 3rd), OMPS data show some AOD perturbations after the first eruption and more significantly elevated values after the second eruption. Like for the Raikoke eruption, WACCM shows immediate and stronger signals during the weeks following the eruptions, but decreasing faster. While for OMPS observations a significant impact (sAOD around 0.01) of the second Ulawun eruption is still apparent in the tropical stratosphere by the end of the year 2019, in the model comparable values are found in October and by the end of 2019 the sAOD has values down to 10 times smaller than for OMPS. The model shows a faster decrease…."
These statements would be aided by the addition of simple line plots of the global and hemispheric sAODs. Figure 7a does show OMPS sAODs integrated over some latitude bands in such a manner, over a longer time period. However, I think that it would be worthwhile indicating the global, 30-90N, 0-30N, 0-30S, etc for both the model and the observations as a comparison.
We believe the global mean sAOD adds valuable information. Therefore, we added another line to Figure 7 (now 8) a and b, the global sAOD (grey). By the way: The latitude bands are not randomly chosen. Raikoke is located at 48 °N. We chose two bands in the NH, North and South of Raikoke, 1 in the tropics and one respectively in the SH.
The Figure label text has been changed accordingly: *'(a) 3- day mean sAOD from OMPS aerosol extinction values (from tropopause altitude up to 30 km) averaged over five latitude ranges (global, 50-90°N: North of Raikoke, 30-50°N: South of Raikoke, 20°S-20°N: tropics and 30-50°S: SH respectively).'*

*Furthermore, we provide a respective Figure in the Appendix (A5) for the WACCM simulation.*

*'Supporting material for Section 4.5:*
*Figure A5: WACCM mean sAOD values for the respective latitude bands, as shown with OMPS and SAGE III/ISS observations in Fig. 8 a and b. When comparing Fig. 8 a and b with Fig. A5 the higher and faster impact on the sAOD from the model simulations become evident (as also shown and explained in Section 4.2).'*

[Figure]

*Figure 1: Respective to Fig. 8 (a) and (b), WACCM means sAOD values. The WACCM AOD is shown here for sulfate only, i.e. with no condensation of water, to eliminate the signature of PSCs in the winter SH, which would likely mask the plume signature closer to the pole.*

We also added a respective sentence in the main text: *'OMPS and SAGE III/ISS data suggest a comparable but smaller sAOD impact for the Australian fires than for the Raikoke eruption (30-50°S compared to 50-90°N).*
*A similar representation of the sAOD as seen in Fig. 8a and b with the WACCM simulation is shown in the supporting material (Fig. A5). Discrepancies in terms of AOD extent and timing, compared to OMPS and SAGE III/ISS observations are also shown in Fig. 5 and explained in Section 4.2.'*

Adding another Figure with slightly changed latitude bands (e.g. 30-90°N, 0-30°N, 0-30°S etc.), does not show new results or give new scientific conclusions. Therefore, we have not added another Figure.

**5) CLAMS model:** The initialisation of the model is pretty coarse (a box) which doesn't have the details of the spatial distribution in the vertical or horizontal within the plume. More care is therefore needed in interpreting the results from the CLAMS model. For example, "Therefore, the sAOD enhancement above Europe observed by OMPS in Fig. 3b does not originate from Raikoke, but rather from forest fires in Alberta, Canada." "Europe" is a large area: The OMPS data suggests that there is an enhancement of the AOD over northern Europe, Scandinavia, the Baltic countries, and western Russia (Fig 3). The CLAMS simulations suggest that the Raikoke plume impacts "southern" Europe. Areas such as the UK at the interface between northern and southern Europe

experience both …… Some of this greater detail is worth stating more explicitly, plus the caveat that the CLAMS initialisation may not be that accurate.

We agree that the text can use more precise details on the location of the Alberta fire plume over Europe, compared to the areas partially impacted from the ClaMS trajectories (Raikoke box). This has been done accordingly: *'At the beginning of July the main bulk of the air mass tracer remains west of the Atlantic Ocean, with only a minimal impact above southern Europe (Fig. 6, second panel). Therefore, the sAOD enhancement above northern Europe observed by OMPS in Fig. 4b does likely originate from other sources than Raikoke (e.g., from forest fires in Alberta, Canada ). not originate from Raikoke, but rather from forest fires in Alberta, Canada.'*

We agree completely that the ClaMS simulation is very simplified in representing the injection location and does not include microphysics and chemistry. This is on the one hand a limitation, because trajectories are calculated which do not correspond to the actual plume (because of the box-shaped initialization area), on the other hand, the simplicity itself is of high value to the study, because we are not analysing quantities, but rather use ClaMS as a pure transport analysis. We try to state even clearer in the revised manuscript that the ClaMS tracer dispersal should be regarded as the effect of pure passive large-scale transport, and that comparison to OMPS allows assessing this effect – but not more. For this please also see answer to Review 1 point 5. Limitations and value of this simplified study are given in the text from line 362 to 370.

**6) Section 4.4.** Vertical distribution. While most of the graphical displays are reasonably logically chosen throughout, here I think that the choice of representation of the vertical distribution could be improved. Figure 6a-d are "around Raikoke and around Ulawan", while Figure 6e shows the OMPS data in a series of time stamps as a function of latitude and altitude. I would have preferred to see the model distributions plotted up in a similar way to the Fig 6e. One could then see if the modelled aerosol plumes interact or overlap (probably more likely) from using either the WACCM model or the CLAMS model. The approximate location of the stratosphere could be marked on Figure 6e and any of the new figures too.

As suggested by the Reviewer, we added a mean tropopause height in Fig. 6e (now Figure 7). Hence, the label description has changed : *'(e) OMPS aerosol extinction monthly averages over all longitudes from June to December 2019. White dashed lines represent the averaged tropopause altitude.'*

Furthermore, we prepared the mentioned model distribution as shown for OMPS in Fig 7e (see below). The main message from this plot is, that the initialization of the injection in the model results in an enhanced aerosol concentration much faster with a subsequent fast transport towards the tropics as well. This is also seen when comparing Figure 4 a and b (now Figure 5) and explained in the text.

Conclusions from the Figure below about a possible interaction/overlapping of the Raikoke and Ulawun plumes are therefore not representative and the Figure was not added to the main text.

[Figure]

*Figure 2: As Figure 7e in the text with WACCM data.*

**7) Section 5: Radiative effects:** "calculated the shortwave RF of the Raikoke and Ulawun plumes using the UVSPEC radiative transfer model (see Sect. 2.7 for the setup of the model and calculations). As input parameters for the model, the SAGE III/ISS volcano-attributed aerosol extinction profiles discussed above are used."

Why isn't the radiative forcing (or the effective radiative forcing) given for the WACCM model? It can be used for these calculations can't it? The use of the SAGE extinction profiles and sensitivity perturbations of the single scattering albedo allow some assessment of the impacts on the clear-sky radiative forcing and the sensitivity to the assumptions. WACCM should be able to give both cloud-free and cloudy sky effective radiative forcings but these are absent from the paper.

How is the surface reflectance taken into account? I could not find details. Won't the co-location of the highest AODs over the highest surface reflectances need to be accounted for (weakening the TOA radiative forcing)?

A possible further study based on radiative forcing (RF) simulations with WACCM has been highly discussed among the authors. There are substantial differences between the model simulation output and satellite observations, especially in terms of aerosol timing and extent (e.g. comparison between Fig. 7 a,b, now Figure 8, and the new Appendix Figure A5). Possible reasons for those discrepancies are given in the main text (i.e. starting from line 330/ last paragraph in section 4.2). SAGE III/ISS and OMPS aerosol extinction values, however, agree very well (e.g. Fig. 7 a and b). RF estimations based on observations (as done in the manuscript in section 5) are therefore more meaningful and are consistent with those derived from a very similar method in Kloss et al. (ACP, 2019) for the 2017 Canadian wildfire plume and in Kloss et al. (JGR, 2020) for the Ambae volcano plume. This gives a similar setup for RF comparisons between these events.

Furthermore, the WACCM simulations in this work have been done by saving and storing chemical and microphysical variables only and not the ones dealing with radiative forcing. Deriving RF information with WACCM requires to calculate and store a large number of corresponding variables all along our WACCM simulations, if we refer to Schmidt et al., 2018 (JGR, 123, 12,491–12,508. https://doi.org/10.1029/ 2018JD028776). For each emission scenario (Raikoke only, Ulawun only, Raikoke+ Ulawun, no volcano emission), we would need to run further independent simulations, i.e. taking all RF effects into account and ignoring aerosol RF effects. This would be a very dedicated work requiring both some new specific expertise with WACCM RF calculations for us and an adapted modelling infrastructure which goes beyond the scope of the presented work. These abilities are expected to be developed in the future. As a result, we have decided to not add any RF calculation with WACCM in the manuscript.

Shortwave surface reflectance was set to a fixed wavelength-independent value of 0.1, which is thought to be a representative value for main surfaces underneath the plume dispersion (ocean, bare and vegetated soil). Of course, the surface reflectance can be substantially higher for ice- and snow-covered surfaces and this adds uncertainties, as now mentioned in the text:

(from L154): '…*(as adopted from the SBDART code). We consider a fixed wavelength-independent value of 0.1 for the surface reflectivity. This is intended to represent an average value for main surfaces underneath the dispersed plume: ocean, bare and vegetated soil. It is important to mention that the surface reflectance can be significantly larger for ice- and snow-covered surfaces; RF estimations can be quite sensitive to the surface reflectance (Sellitto et al., 2016).*'

*Sellitto, P., di Sarra, A., Corradini, S., Boichu, M., Herbin, H., Dubuisson, P., Sèze, G., Meloni, D., Monteleone, F., Merucci, L., Rusalem, J., Salerno, G., Briole, P., and Legras, B.: Synergistic use of Lagrangian dispersion and radiative transfer modelling with satellite and surface remote sensing measurements for the investigation of volcanic plumes: the Mount Etna eruption of 25–27 October 2013, Atmos. Chem. Phys., 16, 6841–6861, https://doi.org/10.5194/acp-16-6841-2016, 2016.*

**Typos/clarifications:**
The level of English is generally acceptable, but there are a number of corrections noted below that will make the paper easier to read and digest. I would suggest that a native English speaker re-read the amended manuscript before re-submission as I won't have caught all of them.
Ok, thank you
l1: stratospheric moderate -> moderate explosive *we avoided the term 'moderate now'*
l15: Suggest Severe -> Explosive ok
l17: of sulfur dioxide (SO2) volcanic emissions -> volcanic emissions of sulphur dioxide (SO2) changed, but differently
l23: dominates -> strongly influences. You cannot say that it dominates as if it were an effusive eruption emitted at the surface it would have littleclimate effect (except perhaps through aerosol-cloud-interactions) ok
l28: Butchart, 2014 -> Butchart, 2014; Jones et al., 2017. I think that the study by Jones et al (2017) is worth including here. Their Figure 1, is perhaps one of the most relevant in terms of the injection latitude and altitude. ok
Jones, A.C., J.M. Haywood, N. Dunstone, M.K. Hawcroft, K. Hodges, A. Jones, and K. Emanuel, Impacts of hemispheric solar geoengineering on tropical cyclone frequency, Nature Communications, 8, 1382, doi:10.1038/s41467-017-01606-0, 2017.
l29: Point (3) does not have a suitable reference associated with it. I would suggest adding the Jones et al (2017) reference again here (see above): relative to the tropopause -> relative to the tropopause (e.g. Jones et al., 2017) ok

l 33: 20Tg SO2 is quite a large estimate for the amount of SO2 injected. I would suggest "Up to around 20Tg SO2" ok

l34: have been -> were ok

l37: climate occurred -> climate has occurred ok

l43: its good practice to be sequential in terms of the dates: Günther et al., 2018; Kristiansen et al., 2010; Krotkov et al., 2010 -> Kristiansen et al., 2010; Krotkov et al., 2010; Günther et al., 2018 true, thank you

l51: the complexity that -> the complexity and the uncertainty that ok

l53: time the -> time, the ok

l54: Canada, Alberta (June) and Siberia (July) -> Alberta, Canada (June) and Siberia, Russia (July) ok

l64: flies -> has flown sentence changed

l85 on multiple -> at multiple: Agreed: is it worth saying explicitly that the wavelength dependence provides information on the aerosol size distribution via the Angstrom exponent?

Yes, the sentence has been changed accordingly:

*"However, the better vertical resolution and observations at multiple wavelengths compared to OMPS, bring an added-value when spatio-temporally averaged data are used for the radiative forcing calculations. The wavelength dependence, for example, can be used to extract information on the aerosol size distribution via the Angstrom exponent."*

l93: to discriminate -> discrimination between ok

104: "volcanic effluents" is a strange phrase: I'd replace with "emitted in volcanic plumes".

The sentence has been changed accordingly*: 'While its primary target is the monitoring of meteorological parameters (surface temperature, temperature, humidity profiles and cloud information), IASI also provides high-quality information on trace gases parameters and particles, including gases and particles emitted by volcanoes (e.g., Clarisse et al., 2013; Carboni et al., 2016; Ventress et al., 2016; Guermazi et al., 2020).'*

L116: micronic -> micron ok

L147: With the UVSPEC the -> With UVSPEC, the ok

L150-l155: remove the "-"s for grammar purposes. *Replaced by numbers*

L191: for a pure -> from a pure *ok*

L212: possibly refer again to Jones et al. (2017) *ok*

L217: as in -> to *ok*

L239: mowing -> moving *ok*

L244 & 247: The use of possibly is questionable. It definitely is converted to sulfate aerosol owing mainly to gas phase oxidation. Remove possibly in both sentences. *ok*

L262: A reference to the smoke from the Alberta fires would be appropriate. There may be better ones appearing at present, but here is one I found: Jenner, L.: Alberta Canada Experiencing an Extreme Fire Season, NASA, May 30, https://www.nasa.gov/image-feature/goddard/2019/ 545 alberta-canada-experiencing-an-extreme-fire-season, 2019. *ok*

Fig 3: Caption – the wavelength for the AOD should be stated. Ok*: 'Global OMPS (at 675 nm) sAOD averaged..'*

L274. even one year -> even nearly one year ok

L279: The eastward transport dominates, which depends on the vertical distribution of the aerosol and the phase of the QBO (quasi-biennial oscillation). The sentence could do with a reference e.g. Lee and Smith, 2003:

Lee, H. and Smith, A.K., 2003. Simulation of the combined effects of solar cycle, quasi-biennial oscillation, and volcanic forcing on stratospheric ozone changes in recent decades. Journal of Geophysical Research: Atmospheres, 108(D2). Ok, thanks

L308: crossed-impact -> cross-impact ok

L325: interfered with the Raikoke evolution -> interfered with the evolution of the Raikoke plume ok

L334: mentioned limitations -> associated limitations ok

L340: which is a schematic estimate, but for sure causes discrepancies compared to observations and reality -> "which is a necessary simplification of reality where pulses in injection altitude and magnitude are inevitable". Better, thanks

L347: potential cloud signatures are included -> cloud signatures are potentially included ok

L357 locationsof -> locations of ok

Section Heading: "Recent" is a subjective term: Kasatochi/Sarychev could be considered to be recent. I would simply add the range of recent to the title "In the context of other recent events (2017-2020)" ok

Fig 7. I like Figure 7. It is very informative. As a minor point, it would have been more logical for the LOAC points to have been plotted in purple so that the latitude of the observations correspond to the latitude band in Fig 7a-b. That has been changed

L451. The slight increase in the observed AOD in April 2019 -> The slight increase in the observed AOD in the southernmost latitude band in April 2019 ok

Fig 8. The 1e-2 scaling on the ordinate axis is tiny! This really needs to be more clear.

This has been done accordingly:

---

## Author Response (AR2)

*We thank Anonymous Reviewer 1 and Jim Haywood for their kind words and suggestions/comments throughout the process, which improved the manuscript. Both points for technical corrections have been considered:*

J.H.: 1) Regarding the CALIOP overpass comment:-
Yes, we are aware of this intersection. With the following Figure it becomes clear that CALIOP did not intersect the bulk of the cloud seen from HIMAWARI, but rather a very thin tail. Therefore, we believe the CALIOP observations not to be of sufficient value for this study.
To my mind, this is a little dismissive. If you are aware of the overpass, then you should state this as it would take only a little modification to the text to signpost this overpass. I suggest:-
"With the exception of an overpass at around 49N that intersected only a narrow tail of the volcanic plume at around 16-17km altitude, there are no CALIOP intersections of the core plume during the early stage."

*Yes, you are right. We have added this sentence.*

J.H.: 2) I am glad that the authors took up the suggestion of adding the global AOD in their Figure 8. However, this does highlight some of the bulk difference between the WACCM model and the observations in terms of the global mean AOD. The background (peak – pre-eruption) for WACCM appears to be 0.025 – 0.0075 (0.00175) while for the OMPS observations we have 0.0125 – 0.005 (0.0075). Generally, the authors do a good job of discussing the limitations of making a like-like comparison between the observations and the modelling (lines 352-375), but it is easy to miss the fact that the background-peak values differ by a factor of ~2.3 in the initial phase. The authors should consider whether they should state this implicitly in the text somewhere around line 352.

*True, we haven't given a qualitative comparison like this so far. You will find a sentence like this now around line 466, after introducing Figure 8 and A5.*